# Exosomes mediate horizontal transmission of viral pathogens from insect vectors to plant phloem

**Qian Chen, Yuyan Liu, Jiping Ren, Panpan Zhong, Manni Chen, Dongsheng Jia, Hongyan Chen, Taiyun Wei\***

Vector-borne Virus Research Center, Fujian Province Key Laboratory of Plant Virology, Institute of Plant Virology, Fujian Agriculture and Forestry University, Fuzhou, China

**Abstract** Numerous piercing-sucking insects can horizontally transmit viral pathogens together with saliva to plant phloem, but the mechanism remains elusive. Here, we report that an important rice reovirus has hijacked small vesicles, referred to as exosomes, to traverse the apical plasmalemma into saliva-stored cavities in the salivary glands of leafhopper vectors. Thus, virions were horizontally transmitted with exosomes into rice phloem to establish the initial plant infection during vector feeding. The purified exosomes secreted from cultured leafhopper cells were enriched with virions. Silencing the exosomal secretion-related small GTPase Rab27a or treatment with the exosomal biogenesis inhibitor GW4869 strongly prevented viral exosomal release in vivo and in vitro. Furthermore, the specific interaction of the 15-nm-long domain of the viral outer capsid protein with Rab5 induced the packaging of virions in exosomes, ultimately activating the Rab27a-dependent exosomal release pathway. We thus anticipate that exosome-mediated viral horizontal transmission is the conserved strategy hijacked by vector-borne viruses.

**\*For correspondence:**
weitaiyun@fafu.edu.cn

**Competing interests:** The authors declare that no competing interests exist.

## Introduction

Many devastating plant, animal, and human viral pathogens are horizontally transmitted by arthropod insects (*Eigenbrode et al., 2018*; *Mayer et al., 2017*). For example, rice stripe virus transmitted by planthoppers is a serious agricultural threat in rice-growing countries throughout Asia (*He et al., 2017*), and Zika virus (ZIKV) transmitted by mosquitoes has been a recent public health threat in the Americas (*Liu et al., 2017*). In general, arthropod-borne viruses (arboviruses) establish their initial infection in the insect midgut, which is disseminated to the hemolymph and ultimately spread into the salivary glands, from which virions are introduced into susceptible hosts together with saliva (*Hogenhout et al., 2008*; *Wei and Li, 2016*). Insect salivary gland cells are filled with abundant apical plasmalemma-lined cavities, where saliva is stored (*Mao et al., 2017*; *Wei and Li, 2016*). Arboviruses have to pass through the apical plasmalemma into insect salivary cavities, thereby moving with salivary flow to establish the initial infection in hosts (*Mayer et al., 2017*; *Wei and Li, 2016*). The secretory cells in the central region of salivary glands of the whitefly have been demonstrated to determine the transmissibility of begomoviruses (*Wei et al., 2014*). However, how arboviruses overcome cavity plasmalemma barriers for successful viral transmission is still poorly understood.

Frequently, arboviruses are sequestered into multivesicular bodies (MVBs) in the salivary glands of insect vectors (*Ammar and Nault, 1985*; *Gray et al., 2014*; *Gray and Gildow, 2003*; *Janzen et al., 1970*; *Mao et al., 2017*; *Shikata and Maramorosch, 1965*). MVBs are spherical endosomal organelles containing small vesicles formed by the inward budding of the limiting membrane into the endosomal lumen (*Chahar et al., 2015*). MVBs fuse with the plasma membrane to release the internal vesicles into the extracellular space in an exocytic manner (*Chahar et al., 2015*). These

released vesicles are known as exosomes, which can transport various cellular regulatory RNAs from cell to cell, thereby receiving extensive research attentions (*Chahar et al., 2015*; *Geisler and Coller, 2013*; *Janas et al., 2015*; *Kim et al., 2017*). Thus, exosome biogenesis starts with the maturation from the early endosomes into late endosomes (*Chahar et al., 2015*). Several proteins are implicated in exosome biogenesis, including Rab GTPases and the tetraspanin CD63 (*Kalluri and LeBleu, 2020*). Endosomal Rab GTPases, such as Rab27a, Rab5, and Rab11, can regulate exosome biogenesis, trafficking, and/or release from cultured animal cells (*Akers et al., 2013*; *Alenquer and Amorim, 2015*; *Zeigerer et al., 2015*). Notably, early endosomal Rab5 is also detected within the MVBs and exosomes (*Logozzi et al., 2009*; *Pisitkun et al., 2004*; *Vidal and Stahl, 1993*). For example, the purified exosomes from hepatitis C virus-infected and uninfected hepatoma cells contain Rab5 (*Ramakrishnaiah et al., 2013*). In *Arabidopsis*, a model plant system, purified exosomes from leaf apoplasts also contain the Rab5 GTPase homologue ARA7 (*Rutter and Innes, 2018*). Indeed, the depletion of Rab5 strongly inhibits exosomal formation and secretion (*Ostrowski et al., 2010*). Furthermore, Rab27a functions in the docking of MVBs in the plasma membrane, thereby regulating exosome secretion (*Ostrowski et al., 2010*). For example, Herpes simplex virus-1 glycoprotein can interact with Rab27a to mediate the exosomal release pathway (*Temme et al., 2010*). Several arboviruses, such as dengue virus (DENV) and ZIKV, hijack exosomes to release virions, as observed in cultured mosquito cells (*Martínez-Rojas et al., 2020*; *Reyes-Ruiz et al., 2019*; *Vora et al., 2018*). Moreover, the secreted exosomes from arbovirus-infected mosquito cells can transmit viruses to cultured mammalian cells (*Martínez-Rojas et al., 2020*; *Reyes-Ruiz et al., 2019*; *Vora et al., 2018*). However, whether exosome-mediated viral spread occurs in infected host tissues or organs in vivo remains unknown. The final bottleneck in viral infection via insect vectors, that is, the cavity plasmalemma of salivary glands, is a principal determinant of the ability of an insect species to transmit a virus (*Hogenhout et al., 2008*; *Wei and Li, 2016*). Accordingly, we can reasonably regard the cavity plasmalemma of insect salivary glands as the plasma membrane of cultured infected cells. Thus, how arboviruses exploit exosomes to overcome the cavity plasmalemma barriers of vector salivary glands is a unique system for us to more easily address the role of viral release via the exosomal pathway in vivo.

Numerous plant viral pathogens of agricultural importance are persistently transmitted by piercing-sucking insect vectors, such as leafhoppers, planthoppers, and whiteflies, to plant phloem (*Hogenhout et al., 2008*; *Wei and Li, 2016*). Rice dwarf virus (RDV), the first recorded plant arbovirus transmitted by leafhopper vectors in 1895, causes a severe rice disease in Asia (*Wei and Li, 2016*). We have previously demonstrated that the plant reovirus RDV can be secreted by cultured leafhopper cells via infectious exosomes (*Wei et al., 2008*; *Wei et al., 2009*). Here, we report that RDV hijacks the exosomal release pathway to traverse the apical plasmalemma into saliva-stored salivary cavities, ultimately moving with exosomes into the phloem of rice plants to establish initial viral infection via leafhopper vectors.

## Results

### RDV traverses the apical plasmalemma into leafhopper salivary cavities via the exosomal release pathway

The salivary glands of the leafhopper *Nephotettix cincticeps* consist of a pair of principal and accessory salivary glands (*Figure 1—figure supplement 1A*). The principal salivary gland (PSG) contains six types of cells (I–VI) (*Figure 1—figure supplement 1A*), which are filled with apical plasmalemma-lined cavities (*Figure 1A* and *Figure 1—figure supplement 1A*). Immunofluorescence microscopy showed that RDV disseminated into the salivary cavities by traversing the apical plasmalemma in all types of PSG cells (*Figure 1—figure supplement 1*). We further used electron microscopy to visualize how RDV disseminated into the salivary cavities. In the cytoplasm of virus-infected type III cells, double-layered RDV particles (approximately 65 nm in diameter) were often engulfed into small vesicles within the MVBs at the periphery of salivary cavities (*Figure 1B and C*). The attachment of MVBs to the cavity plasmalemma drove the enation of the apical plasmalemma to form a membrane extrusion toward the cavity (*Figure 1D and E*). This extrusion was followed by the fusion of the apical plasmalemma of cavities with the plasma membrane of MVBs (*Figure 1F–H*). Finally, virus-containing small vesicles (exosomes) were released into the salivary cavities (*Figure 1I and J*). RDV-

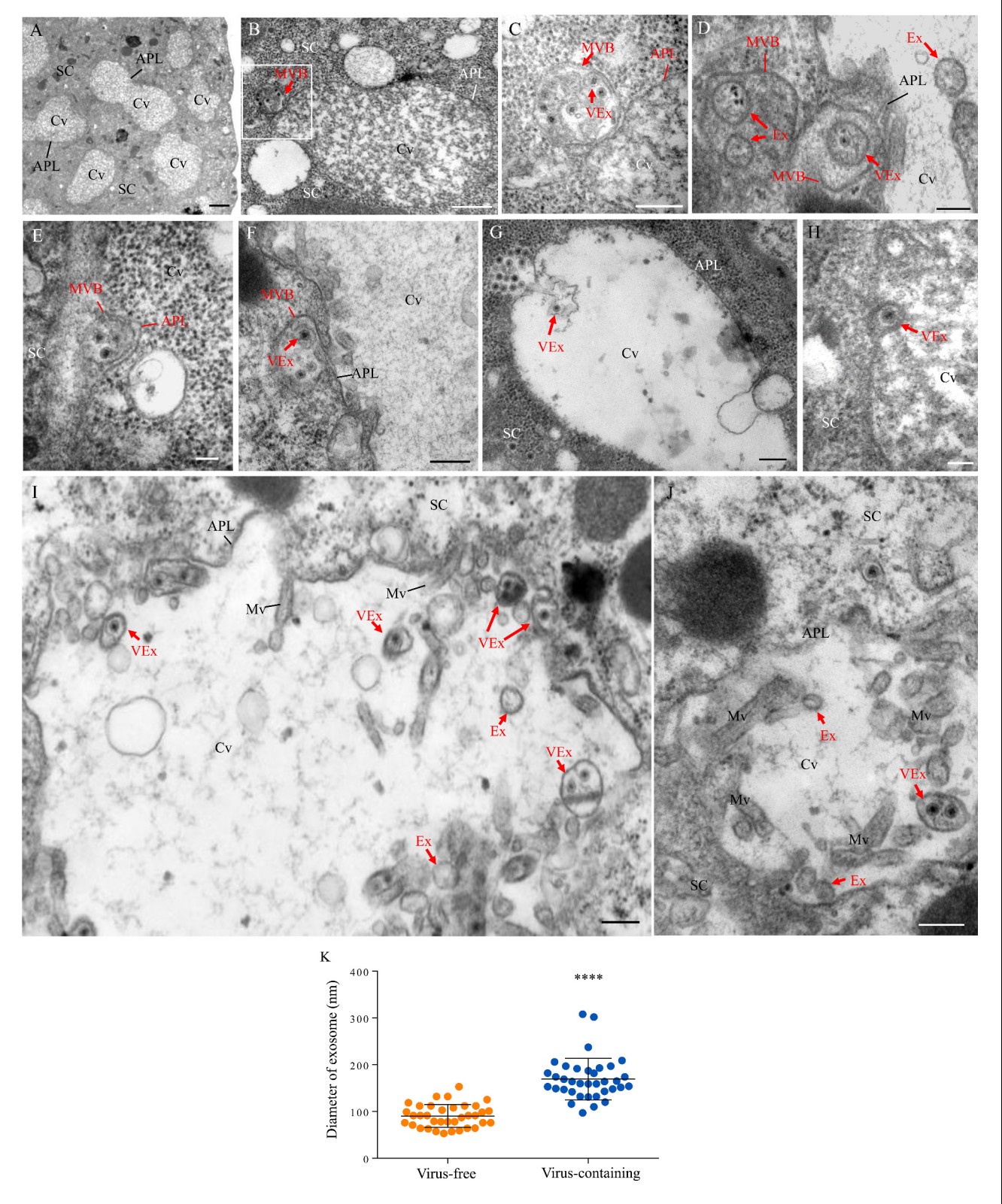

**Figure 1.** Transmission electron micrographs showing that RDV traverses the apical plasmalemma into the salivary cavities via the exosomal release pathway. At 14 days padp, virus-infected salivary glands of *N. cincticeps* were examined by transmission electron microscopy. (**A**) Salivary cavities in type III cells. (**B–C**) Virus-containing MVBs at the periphery of salivary cavities attached to the apical plasmalemma. Panel **C** showing an enlarged image of the boxed area in panel **B**. (**D–E**) The propulsion of MVBs led to the enation of the apical plasmalemma toward the cavity. (**F–H**) Virus-containing

*Figure 1 continued on next page*

Figure 1 continued

MVBs fused with the apical plasmalemma of cavities. (I–J) Virus-containing exosomes were released into the cavities. SC, salivary cytoplasm; APL, apical plasmalemma; Cv, cavity; Ex, exosome; VEx, virus-containing exosome; MVB, multivesicular body; Mv, microvilli. Bars, 2 μm (A), 500 nm (B, F), 200 nm (C, D, G, I and J), and 100 nm (E and H). (K) The mean diameters of exosomes within the salivary cavities, as measured from more than 30 virus-free or virus-containing exosomes. The diameters of exosomes are shown in a dot plot, with the middle line representing the mean value and the top and bottom lines representing the SD. ****p<0.0001.

The online version of this article includes the following figure supplement(s) for figure 1:

**Figure supplement 1.** RDV dissemination into the salivary cavities in the PSG.

**Figure supplement 2.** Immunogold labeling of RDV antigens on viral particles within the MVBs or exosomes in the leafhopper salivary glands or cultured cells.

specific antibody specifically recognized viral particles within the MVBs or exosomes (*Figure 1—figure supplement 2A–C*). Virus-free exosomes measured 53–135 nm in diameter, but the packaging of viral particles enlarged exosomes to 110–302 nm in diameter in the salivary cavities (*Figure 1K*). Together, our observations suggest that RDV exploits an exosomal release pathway to mediate viral entry into the salivary cavities via passage through the apical plasmalemma.

## Viral infection activates the formation of exosomes for the delivery of viral particles to the salivary cavities

Rab27a is an important regulator of exosome release, and the tetraspanin protein CD63 has been described as a key factor in exosomes production (*Kalluri and LeBleu, 2020*; *Ostrowski et al., 2010*). Immunoelectron microscopy indicated that the antibodies against *Drosophila* Rab27a and *N. cincticeps* CD63 can be used to label virus-containing MVBs and exosomes in virus-infected salivary glands (*Figure 2A and B*). Furthermore, these two antibodies also labeled some vesicle-like structures in uninfected salivary glands (*Figure 2A and B*). Immunofluorescence microscopy confirmed that Rab27a or CD63 was colocalized with RDV antigens to form puncta along the cavity plasmalemma or within cavities in virus-infected salivary glands (*Figure 2C and D*). Viral infection induced a significant increase in the average number of Rab27a- or CD63-specific punctate structures in type III cells (*Figure 2E*). RT-qPCR and western blot assays indicated that RDV infection significantly increased the relative expression of Rab27a and CD63 in the salivary glands (*Figure 2F and G*). Thus, RDV infection activated the formation of exosomes for carrying of viral particles. Taken together, these results suggest that viral release from salivary glands into the cavities potentially involves secretory exosomes derived from MVBs.

We then knocked down the expression of Rab27a by RNA interference (RNAi) to confirm the role of exosomes in RDV dissemination from salivary glands. Viruliferous leafhoppers were microinjected with double-stranded RNAs targeting Rab27a or green fluorescence protein (dsRab27a or dsGFP) and then transferred to rice seedlings. RT-qPCR and western blot assays showed that the accumulation levels of the major outer capsid protein P8 of RDV and Rab27a in the salivary glands of dsRab27a-treated insects were significantly decreased compared with in dsGFP controls (*Figure 3A and B*). Immunofluorescence microscopy confirmed that the knockdown of Rab27a expression substantially reduced the accumulation of virus- and Rab27a-positive puncta in the salivary glands, especially within the salivary cavities (*Figure 3C and D*). As expected, dsRab27a treatment significantly decreased the ability of viruliferous leafhoppers to transmit viruses to rice plants (*Figure 3E*). Taken together, these results indicate that the exosomal release of viruses into salivary cavities is greatly impaired by the knockdown of Rab27a expression in leafhoppers, which significantly suppresses the horizontal transmission of RDV from insect salivary glands to rice plants.

## RDV is secreted from cultured insect vector cells via the exosomal release pathway

Previously, we demonstrated that MVBs were involved in the exosomal release of RDV particles from infected cultured leafhopper cells (*Wei et al., 2009*). At 48 hr post-inoculation (hpi) with RDV at a multiplicity of infection (MOI) of 1, electron microscopy showed that double-layered RDV particles were sequestered in the endosomal compartments (*Figure 4A–I*), which then formed MVBs in the cytoplasm of cultured *N. cincticeps* cells (*Figure 4A–II*). Finally, virus-containing exosomes were released from infected cultured cells after the fusion of MVBs with the plasma membrane

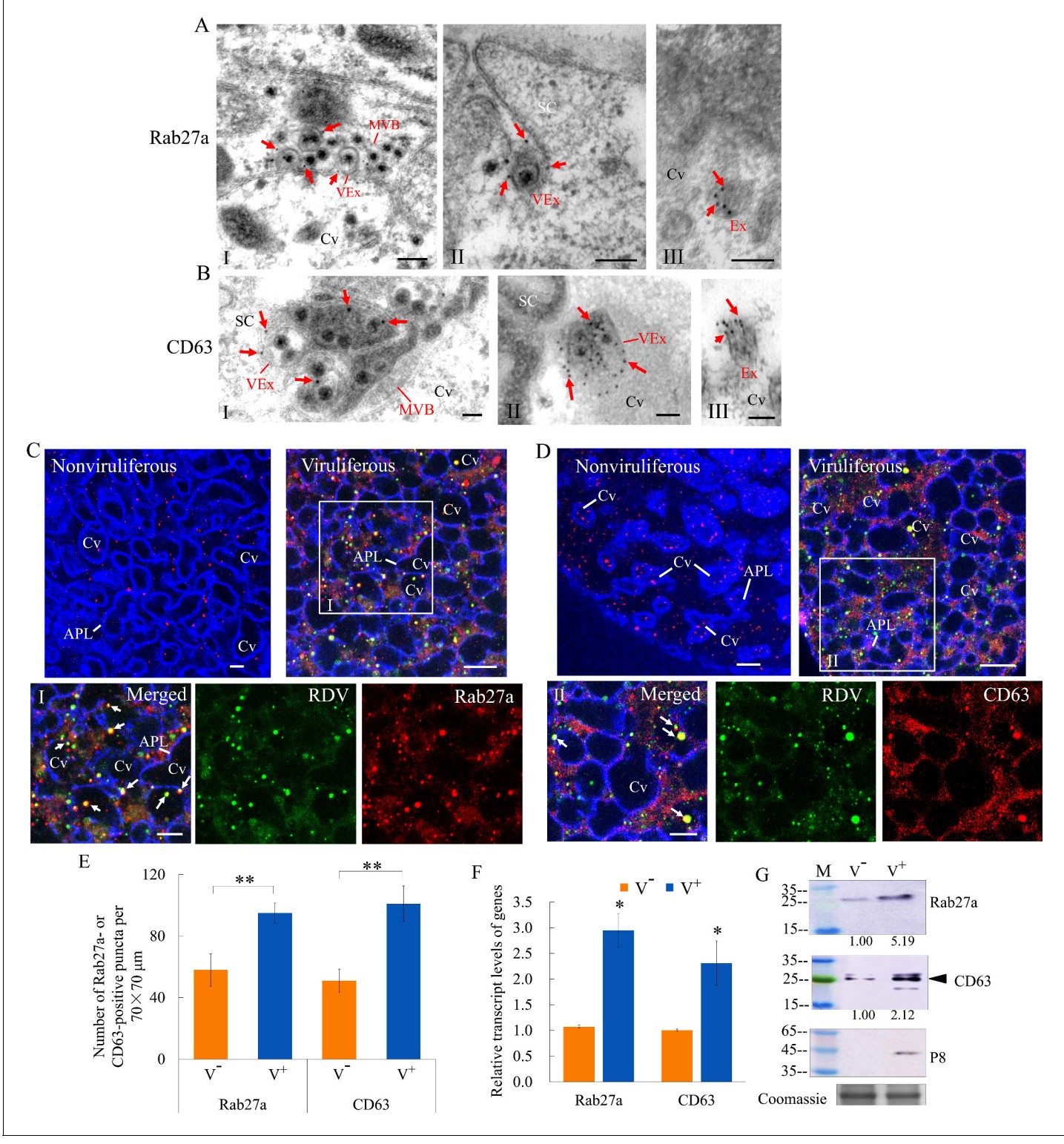

**Figure 2.** RDV dissemination into the salivary cavities via Rab27a-involved exosomal release pathway. (**A–B**) Immunogold labeling of Rab27a (**A**) or CD63 (**B**) on virus-containing MVBs or exosomes in the salivary glands of *N. cincticeps*. Virus-infected (panels I and II) or uninfected (panels III) salivary glands of *N. cincticeps* were immunolabeled with Rab27a-specific IgG or CD63-specific IgG as primary antibody, followed by treatment with 15 nm gold particle-conjugated goat antibody against rabbit IgG as secondary antibody. SC, salivary cytoplasm; MVB, multivesicular body; Cv, cavity; Ex, exosome; VEx, virus-containing exosome. Arrows indicate gold particles. Bars, 100 nm. (**C–D**) Immunofluorescence assay of the distribution of Rab27a (**C**) or CD63 (**D**) during viral infection of the salivary glands of *N. cincticeps*. Salivary glands of nonviruliferous or viruliferous *N. cincticeps* were fixed,

*Figure 2 continued on next page*

Figure 2 continued

immunostained with virus-FITC (green), Rab27a- rhodamine (red) or CD63-rhodamine (red), and actin dye phalloidin-Alexa Fluor 647 carboxylic acid (blue). Then, immunostained salivary glands were processed for immunofluorescence microscopy. Panels I and II (merged) are the enlarged images of the boxed areas in C and D. Arrows show the colocalization puncta in the salivary cavities. APL, apical plasmalemma; Cv, cavity. Bars, 5 μm. (E) The mean number of Rab27a- or CD63-positive puncta in infected or uninfected salivary glands. Twenty random 70 × 70 μm fields of type III cell samples from infected or uninfected salivary glands were examined. (F) RT-qPCR assay showing the transcript levels of Rab27a and CD63 in the salivary glands of viruliferous or nonviruliferous leafhoppers. RT-qPCR results were normalized against the actin expression level, and the transcript levels of Rab27a or CD63 in the salivary glands of nonviruliferous leafhopper were normalized as 1. (G) Western blot assay showing the expression levels of Rab27a and CD63 in the salivary glands of viruliferous or nonviruliferous leafhoppers. The relative intensities of bands in the analyses of Rab27a and CD63 are shown below. Coomassie-blue-stained gels demonstrate the loading of equal amounts of proteins. Data shown here are representative of three biological replicates. Means (± SD) from three biological replicates are shown in E and F. *p<0.05, **p<0.01. V⁻, nonviruliferous; V⁺, viruliferous.

(*Figure 4A–III* and IV), as described previously (*Wei et al., 2008*; *Wei et al., 2009*). Immunoelectron microscopy confirmed that RDV-specific antibody recognized viral particles within exosomes (*Figure 1—figure supplement 2D*). Furthermore, Rab27a and CD63 antibodies reacted with virus-containing exosomes in virus-infected cells and some vesicle-like structures in uninfected cells (*Figure 4B*). Next, we purified exosomes from the extracellular medium of infected or uninfected

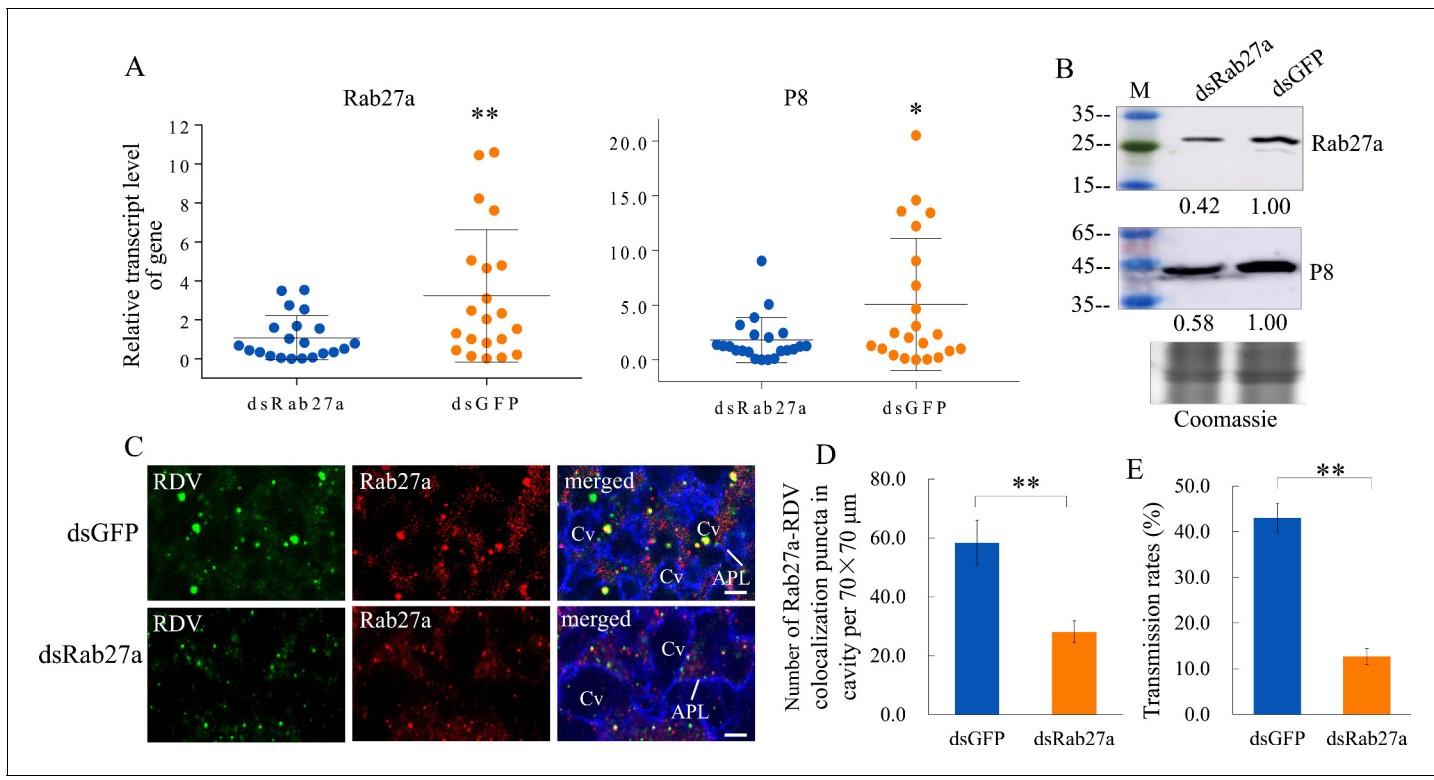

**Figure 3.** Knockdown of Rab27a expression reduces viral dissemination from salivary glands via the exosomal release pathway. (A) RT-qPCR assay showing the transcript levels of RDV P8 and Rab27a expression in the salivary glands of dsRab27a- or dsGFP-treated insects. Results were normalized against the actin expression level. Means (± SD) from three biological replicates are shown. The relative expression levels of the genes in individual insects are shown in a dot plot, with the middle line representing the mean value and the top and bottom lines representing the SD. *p<0.05, **p<0.01. (B) Western blot assay showing the expression levels of RDV P8 and Rab27a in the salivary glands of dsRab27a- or dsGFP-treated insects. The relative intensities of bands in the analyses of Rab27a or RDV P8 are shown below. Coomassie-blue-stained gels demonstrated the loading of equal amounts of protein. Data shown here are representative of three biological replicates. (C) Immunofluorescence microscopy showing the distribution of Rab27a during viral infection in the salivary glands of dsRab27a- or dsGFP-treated insects. Salivary glands were fixed, immunolabeled with virus-FITC (green), Rab27a-rhodamine (red) and actin dye phalloidin-Alexa Fluor 647 carboxylic acid (blue), and processed for immunofluorescence microscopy. APL, apical plasmalemma; Cv, cavity. Bars, 5 μm. (D) The mean number of Rab27a–RDV antigen colocalization puncta in the cavities of dsGFP- and dsRab27a-treated salivary glands. Twenty random 70 × 70 μm fields in type III cell samples from infected or uninfected salivary glands were examined, and means (± SD) from three biological replicates are shown. **p<0.01. (E) The transmission rates by viruliferous dsGFP- or dsRab27a-treated leafhoppers. Means (± SD) from three biological replicates are shown. **p<0.01.

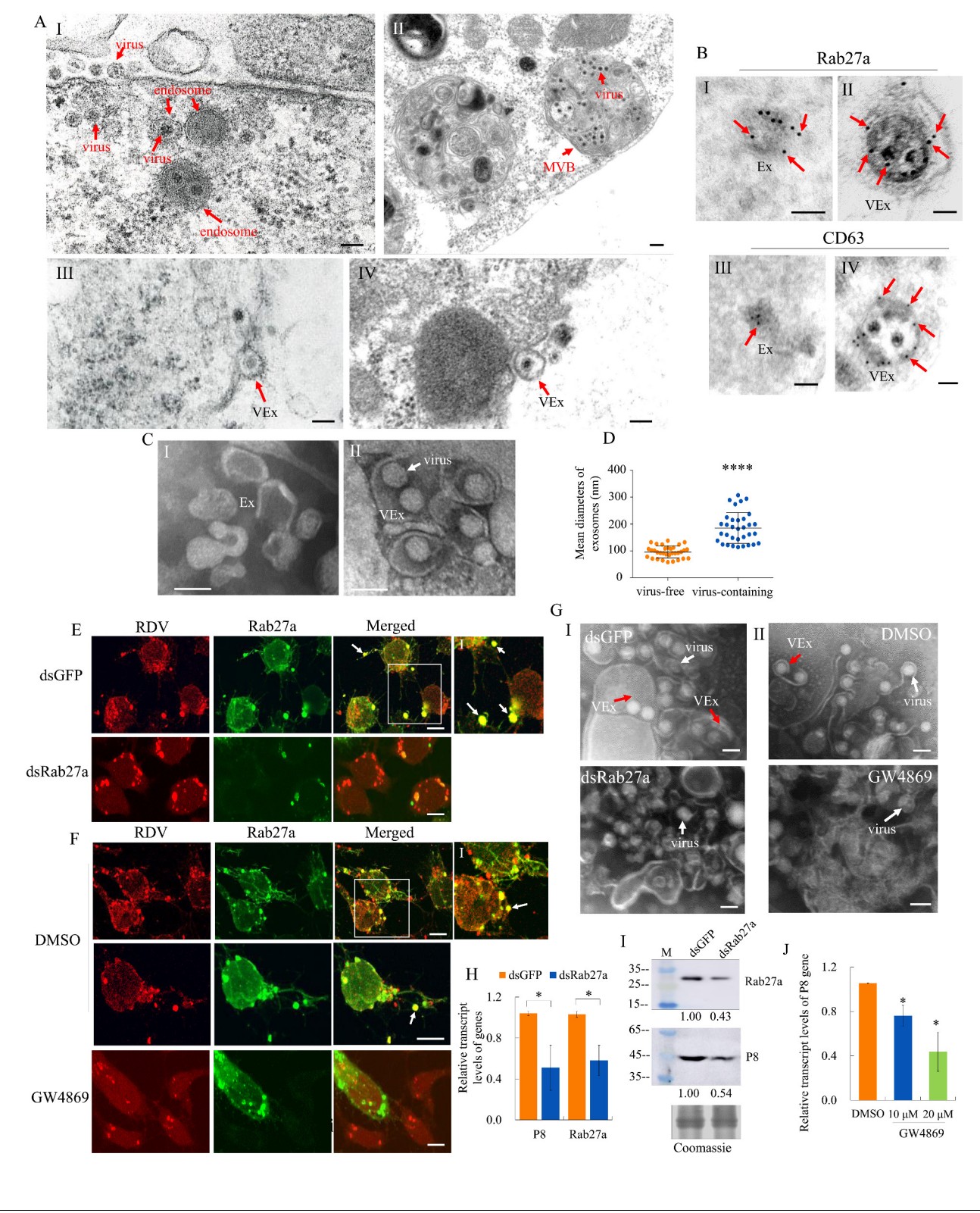

**Figure 4.** The secretion of RDV from cultured insect vector cells via the exosomal release pathway. (**A**) At 48 hpi, virus-infected cultured leafhopper cells were examined by transmission electron microscopy. RDV virions were sequestered in the endosomal compartments (**I**), which then developed into MVBs (**II**). Finally, virus-containing exosomes were secreted from cultured cells (**III-IV**). Bars, 100 nm (**I, III, IV**) and 200 nm (**II**). (**B**) Immunogold labeling of Rab27a or CD63 in virus-free (**I, III**) or virus-containing (**II, IV**) exosomes. Arrows indicate gold particles. At 48 hpi, cultured cells were immunostained

*Figure 4 continued on next page*

*Figure 4 continued*

with Rab27a-specific IgG or CD63-specific IgG as primary antibody, followed by treatment with 15 nm gold particle-conjugated goat antibody against rabbit IgG as secondary antibody. Bars, 100 nm. (C) Negative staining electron microscopy showing the purified exosomes from the extracellular media of uninfected (I) or virus-infected (II) cultured cells. Bars, 100 nm. (D) The diameters of purified exosomes are shown in a dot plot, with the middle line representing the mean value and the top and bottom lines representing the SD. ****$p<0.0001$. (E–F) Virus-infected cultured cells treated with dsRab27a (E) or 20 µM GW4869 (F) were immunostained with virus-rhodamine (red) and Rab27a-FITC (green) and examined by immunofluorescence microscopy. Treatments with dsGFP (E) or DMSO (F) served as the controls. Arrows indicate the colocalization puncta of virus-rhodamine and Rab27a-FITC on the cell surface or outside of cells. Panels I show the enlarged images of the boxed areas in panels E or F. Bars, 5 µm. (G) Negative staining electron microscopy showing the purified exosomes from the extracellular media of dsGFP-, dsRab27a-, DMSO-, or 20 µM GW4869-treated virus-infected cultured cells. Panel I, dsGFP- and dsRab27a-treated cells; Panel II, DMSO- and GW4869-treated cells. Bars, 100 nm. (H) RT-qPCR assay showing the transcript levels of RDV P8 and Rab27a in dsRab27a- or dsGFP-treated cells. Results were normalized against the actin transcript level; the expression levels of Rab27a or P8 in dsGFP-treated cells were normalized as 1. Means (± SD) from three biological replicates are shown. *$p<0.05$. (I) Western blot assay showing the expression levels of RDV P8 and Rab27a in dsRab27a- or dsGFP-treated cells. The relative intensities of bands in the analyses of Rab27a and P8 are shown below. Coomassie-blue-stained gels demonstrated the loading of equal amounts of protein. Data shown here are representative of three replicates. (J) RT-qPCR assay showing the transcript levels of RDV P8 in GW4869 (10 or 20 µM)- or DMSO-treated cells. Means (± SD) from three biological replicates are shown. *$p<0.05$. Ex in A–C, exosome; VEx in A–C and G, virus-containing exosome; MVB in A, multivesicular body.

The online version of this article includes the following figure supplement(s) for figure 4:

**Figure supplement 1.** Effects of treatment with dsRab27a, dsGFP, GW4869 (20 µM), or DMSO on the accumulation of cell-associated and extracellular viruses in cultured leafhopper cells infected with RDV (MOI of 10) at 48 hpi.

cultured leafhopper cells. Negative staining electron microscopy showed that virus-containing exosomes (114–307 nm in diameter) can be isolated from the extracellular medium of infected cells, which were thereby larger than virus-free exosomes (58–138 nm in diameter) secreted from uninfected cells (*Figure 4C and D*).

GW4869 is a cell-permeable but selective inhibitor for exosome production and release (*Vora et al., 2018*). Immunofluorescence microscopy showed that treatment with dsRab27a or 20 µM GW4869 followed by RDV infection (MOI of 1) efficiently inhibited virus-containing exosome formation and release (*Figure 4E and F*). Negative staining electron microscopy of purified exosomes secreted from virus-infected cells further confirmed that treatment with dsRab27a or GW4869 efficiently abolished exosome formation (*Figure 4G*). RT-qPCR or western blot assays showed that the accumulation levels of RDV P8 and Rab27a in dsRab27a-treated cultured cells were substantially decreased compared with that in dsGFP-treated controls (*Figure 4H and I*). Treatment of leafhopper cells with 20 µM GW4869 also significantly reduced RDV P8 transcript accumulation, compared with the DMSO control (*Figure 4J*).

Our previous study showed the RDV particles were assembled at the periphery of the viroplasm, the site of viral replication and assembly, and then engulfed by MVBs in virus-infected cultured leafhopper cells (*Wei et al., 2006b*; *Wei et al., 2008*; *Wei et al., 2009*). We examined whether the treatment with dsRab27a or GW4869 had any effects on the replication of RDV in cultured cells. The cultured cells treated with dsRNAs, DMSO, or 20 µM GW4869 were inoculated with purified RDV at a MOI of 10 which guaranteed viral infection rate was 100%. At 48 hpi, the extracellular medium and the cells were collected. Treatment with dsRab27a or GW4869 significantly reduced RDV P8 transcript loads in the extracellular medium, but did not significantly reduce RDV P8 transcript loads in cell-associated viruses (*Figure 4—figure supplement 1*). These results demonstrated that RDV normally proliferated in the infected cells, but viral release from the cells was impeded by the inhibition of exosome formation. Taken together, these results indicate that RDV can be released from cultured insect vector cells via the Rab27a-dependent exosomal release pathway.

## RDV P2 inherently interacts with Rab5 to mediate the exosomal release of virions

We then investigated how RDV virions were sequestered by exosomes. The outer layer of RDV virion contains the major outer capsid protein P8 and minor outer capsid protein P2 (*Miyazaki et al., 2016*). Structural analysis shows that P2 has an L-shaped flexible structure with a 10-nm-long domain anchored on the viral surface (P2C) and a 15-nm-long domain oriented toward the exterior (P2N), which recognizes cellular receptors to mediate viral entry into insect vector cells (*Miyazaki et al., 2016*; *Figure 5A*). Rab5 is localized on the membrane surface of both early and mature endosomal

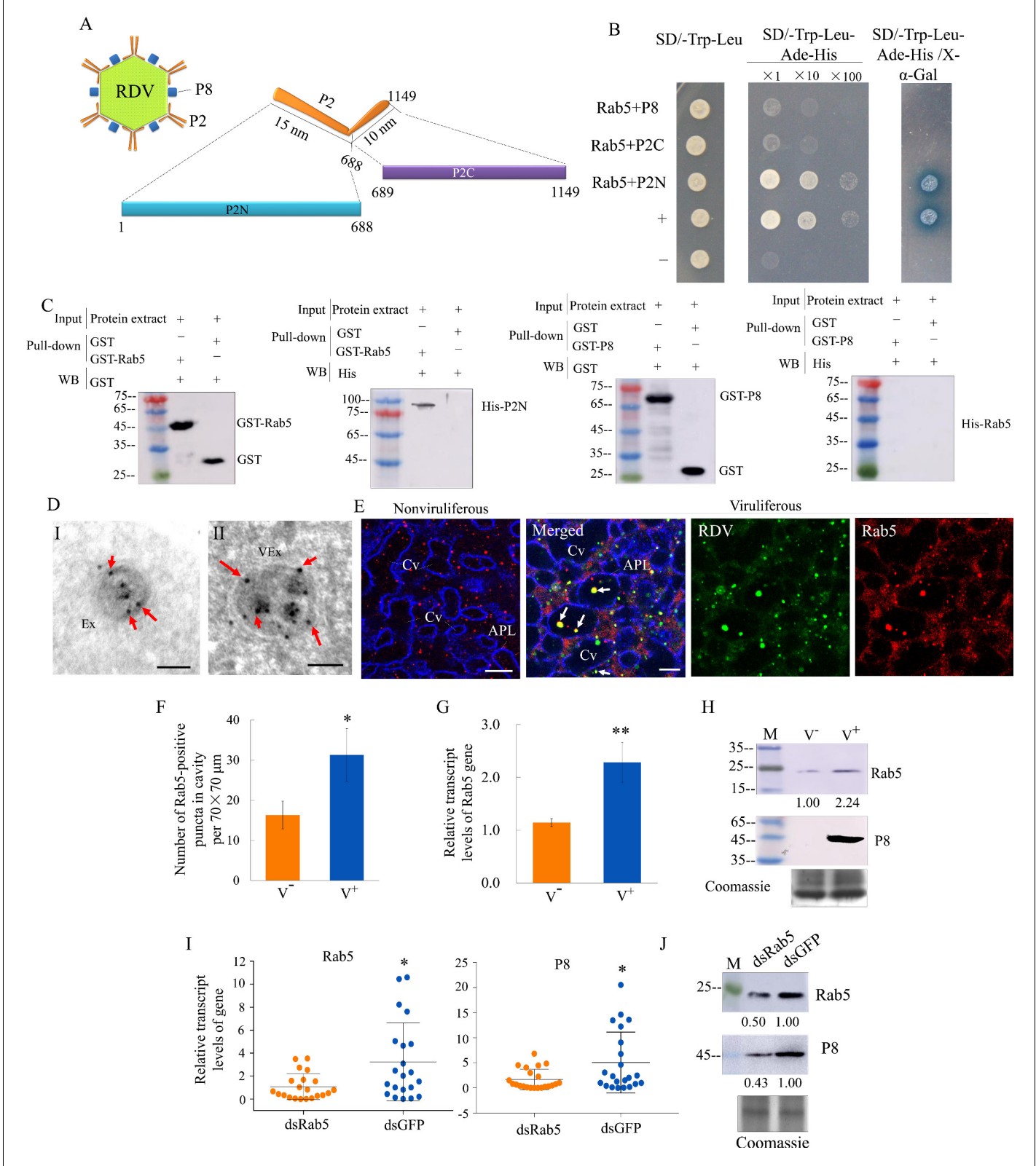

**Figure 5.** RDV P2 interacts with endosomal Rab5 of leafhopper. (**A**) Schematic representation of the structures of RDV particle and P2 protein. P2 is composed of an L-shaped flexible structure with a 15 nm domain (aa 1–688, P2N) toward the exterior, and a 10 nm domain (aa 689–1149, P2C) on the viral envelope. (**B**) RDV P2N, but not RDV P2C or P8, specifically interacted with Rab5 in a yeast two-hybrid assay. Transformants on SD/-Trp-Leu-Ade-His plates are labeled as follows: +, positive control, i.e., pGBKT7-53/pGADT7-T; –, negative control, i.e., pGBKT7-Lam/pGADT7-T; Rab5 +P8,

*Figure 5 continued on next page*

*Figure 5 continued*

pGADT7-Rab5/pGBKT7-P8; Rab5 +P2C, pGADT7-Rab5/pGBKT7-P2C; Rab5 +P2N, pGADT7-Rab5/pGBKT7-P2N. Yeast cultures appeared blue in the β-galactosidase assay. (C) GST pull-down assay demonstrating the interaction of P2N with Rab5. Rab5 or RDV P8 fused with GST served as the bait, and GST alone served as the control. P2N or Rab5 fused with His served as the prey. The bait protein or the GST control were incubated with cell lysate expressing the His-fused protein. Input and pull-down samples were probed with antibodies against GST or His for western blot assays. (D) Immunogold labeling of Rab5 on exosomes in the salivary cavities. The salivary glands from nonviruliferous (panel I) or viruliferous (panel II) leafhoppers were immunostained with Rab5-specific IgG as primary antibody, followed by treatment with 15 nm gold particle-conjugated goat antibody against rabbit IgG as secondary antibody. Arrows indicate gold particles. Ex, exosome; VEx, virus-containing exosome. Bars, 100 nm. (E) Immunofluorescence assay showing the distribution of Rab5 during virus infection in the salivary glands of *N. cincticeps*. Virus-infected or uninfected salivary glands of *N. cincticeps* were fixed, immunostained with virus-FITC (green), Rab5-rhodamine (red), and actin dye phalloidin-Alexa Fluor 647 carboxylic acid (blue). Immunostained salivary glands were then processed for immunofluorescence microscopy. Arrows show the colocalization puncta in the salivary cavities. APL, apical plasmalemma; Cv, cavity. Bars, 10 μm. (F) The mean number of Rab5-positive puncta in infected or uninfected salivary glands. Twenty random 70 × 70 μm fields of type III cell samples from infected or uninfected salivary glands were examined. (G) RT-qPCR assay showing the transcript levels of Rab5 in the salivary glands of viruliferous or nonviruliferous leafhoppers. RT-qPCR results were normalized against the actin expression level, and the transcript levels of Rab5 in the salivary glands of nonviruliferous leafhopper were normalized as 1. Means (± SD) from three biological replicates are shown in **F** and **G**. *p<0.05; **p<0.01. (H) Western blot assay showing the expression levels of Rab5 in the salivary glands of viruliferous or nonviruliferous leafhoppers. The relative intensities of bands in the analyses of Rab5 and RDV P8 are shown below. Coomassie-blue-stained gels demonstrated the loading of equal amounts of proteins. Data shown here are representative of three biological replicates. (I) RT-qPCR assay showing the transcript levels of RDV P8 and Rab5 in the salivary glands of dsRab5- or dsGFP-treated insects. Results are normalized against the actin expression level. The relative expression levels of the genes in individual insects are shown in a dot plot, with the middle line representing the mean value and the top and bottom lines representing the SD. *p<0.05. (J) Western blot assay showing the expression levels of RDV P8 and Rab5 in the salivary glands of dsRab5- or dsGFP-treated insects. The relative intensities of bands are shown below. Coomassie-blue-stained gels demonstrated the loading of equal amounts of protein. Data shown here are representative of three replicates. V⁻, nonviruliferous insects; V⁺, viruliferous insects.

The online version of this article includes the following figure supplement(s) for figure 5:

**Figure supplement 1.** Bioinformatic analyses of Rab5 of *N. cincticeps*.

**Figure supplement 2.** Bioinformatic analyses of Rab27a of *N. cincticeps*.

**Figure supplement 3.** Basic Local Alignment Search Tool of Protein Databases (BLASTP) showing the similarity of Rab27a (A) or Rab5 (B) from *N. cincticeps* with homologues from *Drosophila*.

---

MVBs (*Gorvel et al., 1991*; *Mercer et al., 2010*). The open-reading frames (ORFs) of *N. cincticeps* Rab5 and Rab27a were 645 and 672 bp long, respectively, and encoded approximately 23 kDa and 25 kDa predicted proteins, respectively, with the typical Rab family motifs (*Figure 5—figure supplement 1A* and *Figure 5—figure supplement 2A*). Each gene sequence was deposited in GenBank (accession numbers MW266984 and MW266985, respectively). Phylogenic analysis showed that the amino acid sequences of *N. cincticeps* Rab5 and Rab27a clustered with homologs in the order Hemiptera (*Figure 5—figure supplement 1B* and *Figure 5—figure supplement 2B*). Yeast two-hybrid assays showed that RDV P2N, but not RDV P2C or RDV P8, specifically interacted with Rab5 (*Figure 5B*). However, P2N did not interact with Rab27a from *N. cincticeps* (*Figure 5—figure supplement 2C*). Glutathione *S*-transferase (GST) pull-down assays confirmed these interactions (*Figure 5C*). Immunoelectron microscopy indicated that the antibody against *Drosophila* Rab5 can be used to label virus-containing exosomes in virus-infected salivary glands (*Figure 5D*). Notably, Rab5 antibody could react with viral particles within exosomes (*Figure 5D*). Furthermore, this antibody can also label some vesicle-like structures in uninfected salivary glands (*Figure 5D*). Immunofluorescence microscopy confirmed that Rab5 was colocalized with RDV antigens to form puncta within the salivary cavities (*Figure 5E*). RDV infection induced the accumulation of virus- and Rab5-positive puncta in the salivary glands, especially within the salivary cavities (*Figure 5F*). RT-qPCR and western blot assays indicated that RDV infection significantly increased the expression of Rab5 in the salivary glands (Figure G and H). Furthermore, RT-qPCR and western blot assays showed that the accumulation levels of RDV P8 and Rab5 in the salivary glands of dsRab5-treated insects were significantly decreased compared with that in dsGFP controls (*Figure 5I and J*). RT-qPCR assay also showed that the knockdown of Rab5 expression did not significantly affect Rab27a expression and that the knockdown of Rab27a expression also did not significantly affect Rab5 expression (*Figure 5—figure supplement 2D*). However, immunofluorescence microscopy showed that some Rab5-labeled punctate structures were colocalized with Rab27a-labeled punctate structures in the salivary glands (*Figure 5—figure supplement 2E*). Taken together, we deduce that the packaging of RDV into exosomes is caused by the direct interaction of RDV P2N with Rab5.

To further determine how RDV P2N recognizes endosomal MVBs, we infected cultured *Spodoptera frugiperda* (Sf9) cells with recombinant baculoviruses overexpressing His-fused P2N or Myc-fused Rab5. When expressed alone, P2N formed punctate structures, while the overexpressed Rab5 was diffusely distributed in the cytoplasm of Sf9 cells (*Figure 6A*). The coinfection led to the redistribution of Rab5 into the punctate structures of P2N (*Figure 6A*). Notably, antibodies against *Drosophila* Rab5 or Rab27a could also be used to label the endosome-like compartments in the cytoplasm of Sf9 cells (*Figure 6B*), as previously reported for cells of other species (*Nielsen et al., 1999*; *Simonsen et al., 1998*; *Wucherpfennig et al., 2003*). We found that the expressed P2N was colocalized with Rab5- or Rab27a-labeled endosome-like compartments (*Figure 6B*). In particular, P2N-specific endosome-like compartments can release into the extracellular medium of Sf9 cells (*Figure 6B*). Treatment with GW4869 (20 μM) or chloroquine (280 μM) significantly prevented the formation of P2N-specific endosome-like compartments (*Figure 6C* and *Figure 6—figure supplement 1*). Furthermore, the GFP-fused P2N (P2N-GFP) in Sf9 cells also appeared as punctuate inclusions (*Figure 6—figure supplement 2A–B*). Treatment with chloroquine (280 μM) or GW4869 (20 μM) also significantly reduced the mean of fluorescence intensity and accumulation of P2N-GFP (*Figure 6—figure supplement 2C* to E). Thus, RDV P2N had the inherent ability to recognize endosomes, even in the absence of viral infection. Taking together, these results indicate that the specific and inherent interaction of RDV P2 with Rab5 mediates the packaging of RDV particles into exosomes.

## RDV is released from the salivary glands into rice phloem with the exosomes

Generally, viruses in the salivary cavities move with salivary flow to the canal in the stylets and ultimately are injected into the sieve cells of plant phloem as piercing-sucking insect vectors feed on susceptible hosts (*Hogenhout et al., 2008*; *Wei and Li, 2016*). We then tested whether RDV-containing exosomes in the salivary cavities could be ejected with saliva into rice plants. Immunofluorescence assays showed that the antibodies against Rab5, Rab27a, or CD63 did not react with any specific structures in rice phloem (*Figure 7A*). However, Rab5, Rab27a, or CD63 appeared in the sieve tube cells of rice phloem after being fed on by nonviruliferous leafhoppers for 2 days (*Figure 7B*), confirming that the exosomes were released with salivary flow into plant phloem via leafhopper feeding. Notably, RDV was accompanied by Rab5-, Rab27a-, or CD63-involved exosomes into the sieve tube cells of rice phloem after viruliferous leafhoppers feeding and then spread through rice phloem to establish initial infection (*Figure 7C*). Western blot assay further showed that more Rab5, Rab27a, and CD63 proteins were detected in rice seedlings that had been fed on by viruliferous leafhoppers for 2 days compared to those observed after feeding by nonviruliferous leafhoppers (*Figure 7D*). More importantly, western blot assay confirmed that the antibodies against Rab5, Rab27a, or CD63 did not react with the components in rice phloem (*Figure 7D*). Together, these results suggest that RDV is disseminated together with secretory exosomes from insect vector salivary glands into rice phloem, ultimately accomplishing viral transmission to the plant host.

## Discussion

The horizontal transmission of arboviruses from the salivary glands of insect vectors to susceptible hosts is an essential process for viral survival in nature. Through horizontal transmission, infectious virions are ejected into hosts by insect stylets and then robustly establish the initial infection in hosts. Exosome-mediated spread of virions and various cellular or viral regulatory RNAs from cell to cell has gained considerable interest among researchers (*Chahar et al., 2015*; *Geisler and Coller, 2013*; *Janas et al., 2015*; *Kim et al., 2017*; *Mori et al., 2008*; *Tanggis et al., 2017*; *Nguyen et al., 2003*); however, these studies have been limited to in vitro systems of cultured animal cells. Here, we show that the specific interaction of the 15-nm-long domain of RDV P2 on viral surface with Rab5 on the cytoplasmic face of endosomes induces the packaging of intact virions into endosomal MVBs in salivary glands. Such MVBs attach to the apical plasmalemma and induce an exocytosis process promoting the release of virus-containing exosomes into salivary cavities. The average diameters of the salivary canal of leafhoppers in Cicadellidae range from 3 μm to 500 nm (*Brozek and Herczek, 2000*; *Leopold et al., 2003*; *Zhao et al., 2010*), greatly exceeding the maximum diameter of virus-containing exosomes. Therefore, such virus-containing exosomes can move easily through the

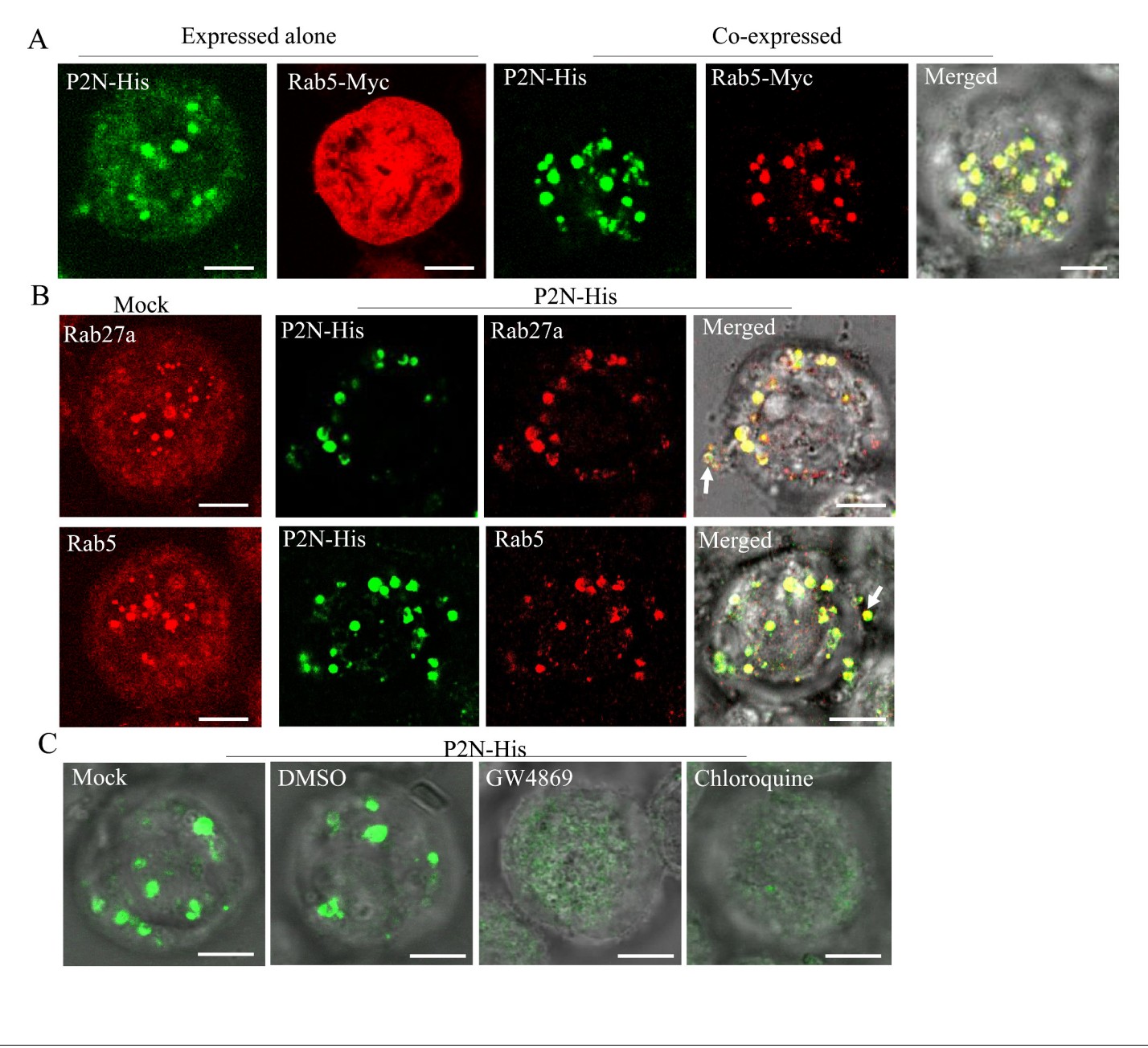

**Figure 6.** RDV P2N is associated with the endosomal compartments in Sf9 cells. (**A**) Immunofluorescence assay showing the colocalization of RDV P2N with Rab5 of *N. cincticeps* in Sf9 cells. RDV P2N-His and Rab5-Myc were expressed alone or co-expressed. Sf9 cells were fixed at 48 hpi and immunolabeled with His-Alexa Fluor 488 (green) and Myc-Alexa Fluor 555 (red). (**B**) The association of RDV P2N with the endosomal compartments labeled by Rab5- or Rab27a-specific IgG in Sf9 cells. The control or P2N-expressing cells were immunostained with Rab5- or Rab27a-rhodamine and His-Alexa Fluor 488. Arrows indicate the release of P2N-specific endosomal compartments into the extracellular medium. (**C**) Endosomal inhibitors inhibited the association of P2N with the endosomal compartments. Sf9 cells were infected with recombinant baculovirus encoding P2N-His. At 48 hpi, cells were treated with GW4869 (20 μM) or chloroquine (280 μM). Cells were immunostained with His-Alexa Fluor 488 prior to examination by immunofluorescence microscopy. Bars, 5 μm.

The online version of this article includes the following figure supplement(s) for figure 6:

**Figure supplement 1.** The effects of chloroquine or GW4869 on the expression of P2N-His in Sf9 cells.

**Figure supplement 2.** The effects of chloroquine or GW4869 on the expression of P2N-GFP in Sf9 cells.

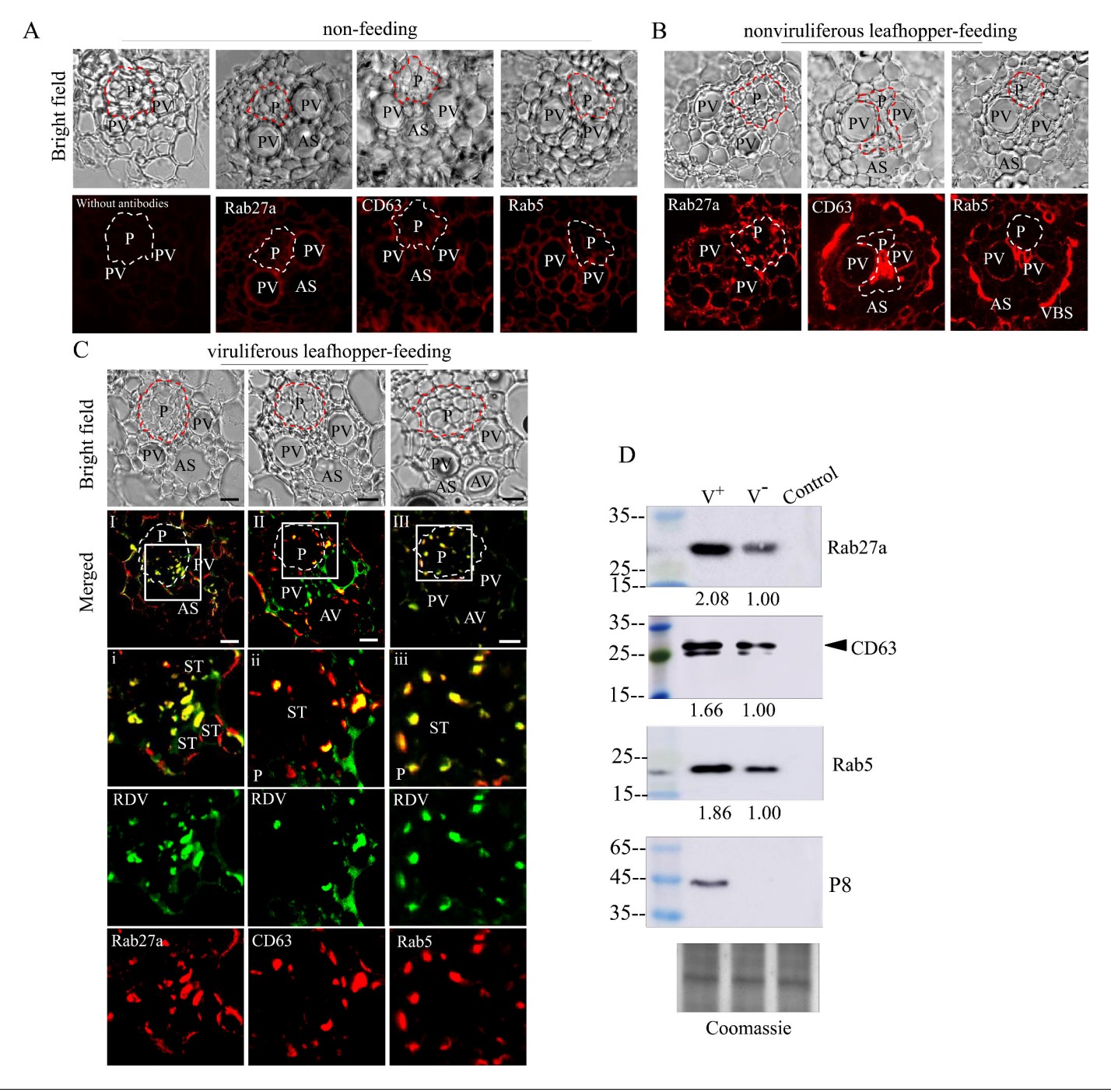

**Figure 7.** Exosome-mediated RDV delivery to the phloem of rice plants via leafhopper feeding. (**A–C**) Immunofluorescence microscopy showing the distribution of RDV antigens, Rab27a, CD63 or Rab5 in the phloem of rice plants after 2 day feeding of viruliferous or nonviruliferous leafhoppers. Rice plants without leafhopper feeding (**A**) and fed on by nonviruliferous (**B**) or viruliferous (**C**) leafhoppers were immunolabeled with virus-FITC (green) and Rab27a, CD63, or Rab5-rhodamine (red) and then examined by immunofluorescence microscopy. Panels i to iii are enlarged images of the boxed areas in panels I to III, respectively. The green and red panels show the separate green (RDV antigens) or red fluorescence (Rab27a, CD63, or Rab5 antigens) of the merged images in panels i to iii. Areas enclosed with a dotted line indicate phloem. AS, air space; P, phloem; PV, pitted vessel; AV, annular vessel; ST, sieve tube; VBS, vascular bundle sheath. Bars, 10 μm. (**D**) Western blot assay of RDV P8, Rab27a, CD63, or Rab5 in rice plants that had been fed on by viruliferous or nonviruliferous leafhoppers. Samples were separated by SDS-PAGE and detected with P8-, Rab27a-, CD63-, or Rab5-specific antibody. V⁻, rice plants that had been fed on by nonviruliferous leafhoppers; V⁺, rice plants that had been fed on by viruliferous leafhoppers; Control, rice plants without leafhopper feeding. The relative intensities of bands in analyses of Rab5, Rab27a, and CD63 are shown below. Coomassie-blue-stained gels demonstrated the loading of equal amounts of protein. Data shown here are representative of three biological replicates.

*Figure 7 continued on next page*

*Figure 7 continued*

The online version of this article includes the following figure supplement(s) for figure 7:

**Figure supplement 1.** Schematic illustration of the feeding cage used for collecting components released from leafhopper salivary glands.

salivary canal. Finally, the virus can be horizontally transmitted with exosomes into rice phloem to establish the initial infection. Inhibition of exosome secretion by silencing Rab5 or Rab27a can prevent such viral exosomal release into salivary cavities, ultimately suppressing viral transmission by insect vectors. More importantly, viral infection upregulates the expression of Rab5 or Rab27a, thereby activating the formation of exosomes for delivery of viral particles to the salivary cavities. Accordingly, the packaging of virions enlarges the normal sizes of exosomes secreted by infected salivary glands cells. Thus, the virus has evolved strategies to activate and exploit the exosomal release pathway into the salivary cavity, facilitating viral transmission by leafhopper vectors to rice phloem together with saliva (*Figure 8*).

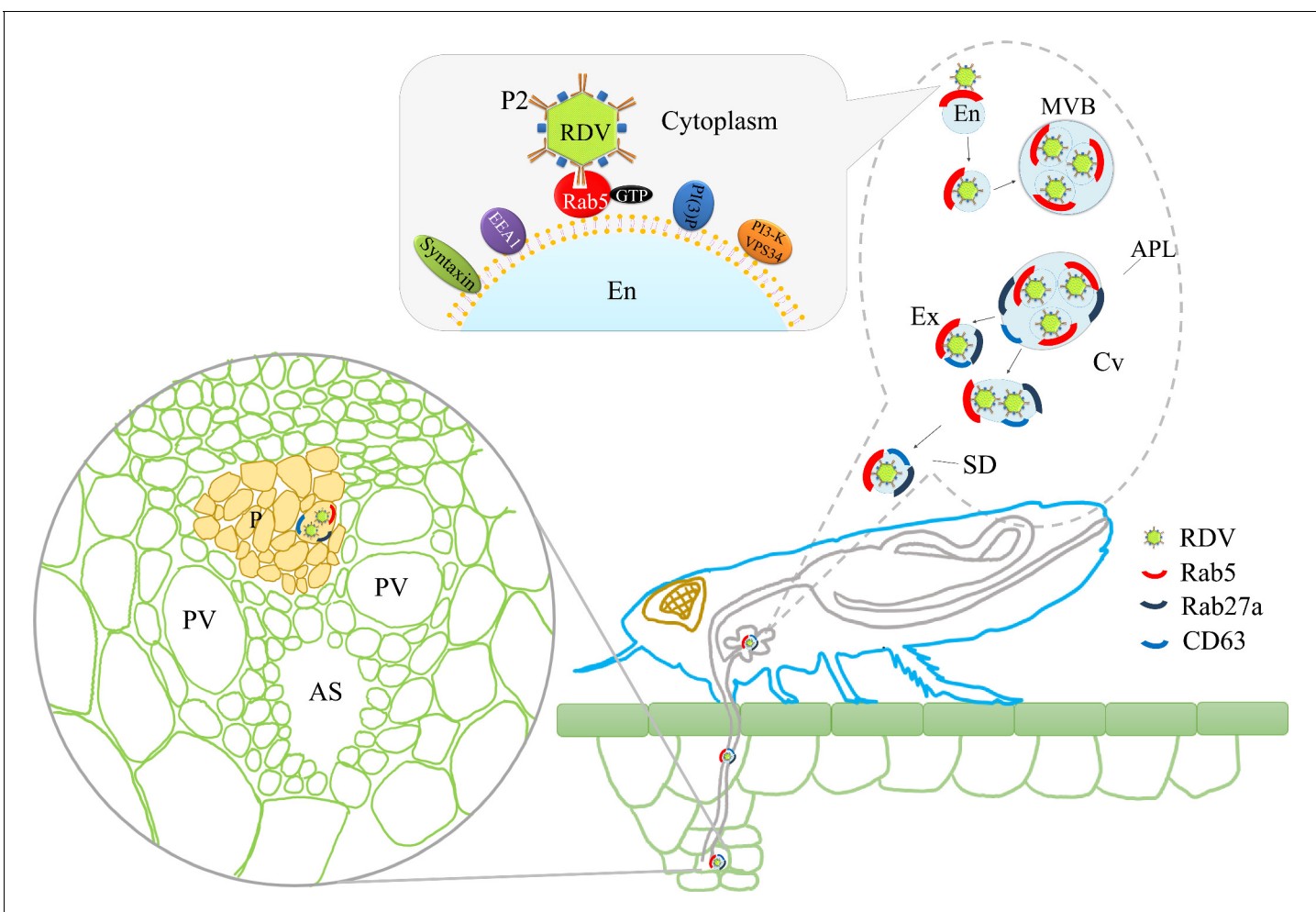

**Figure 8.** Proposed model of RDV hijacking the exosomal release pathway to be released from salivary glands of leafhopper vectors into rice phloem. In the proposed model, the specific interaction of RDV P2N and Rab5 causes the internalization and delivery of viruses to endosomes. Then, these virus-containing endosomes are engulfed in the MVBs in the cytoplasm. The attachment of MVBs with the apical plasmalemma drives the enation of the apical plasmalemma toward the cavity. The fusion of MVBs with apical plasmalemma leads to the release of virus-containing exosomes to the cavities. This exosomal release pathway is mediated by Rab27a. Exosomes are secreted with the salivary flow from the salivary cavities into the stylets and are then injected into the plant phloem as leafhoppers feed on rice plants. The virus released into the phloem establishes the initial replication within plant cells. APL, apical plasmalemma; MVB, multivesicular body; Cv, cavity; SD, salivary duct; En, endosome; Ex, exosome; P, phloem; AS, air space; PV, pitted vessel.

We have shown that the 15-nm-long domain of RDV P2 can interact with cellular receptors on the plasma membrane of cultured leafhopper cells to facilitate viral entry, or with the outer membrane protein of the obligate bacterial symbiont *Sulcia* of leafhoppers to facilitate viral transovarial transmission (*Jia et al., 2017*; *Miyazaki et al., 2016*; *Wei et al., 2007*; *Zhou et al., 2007*). Thus, RDV P2 has evolved to interact with distinct receptors in insect bodies to enable the virus to overcome various tissue or organ barriers. RDV enters insect vector cells through receptor-mediated, clathrin-dependent endocytosis and is sequestered in endosomal compartments (*Wei et al., 2007*). Furthermore, RDV P2 can induce membrane fusion in insect vector cells (*Zhou et al., 2007*). We also show that the 15-nm-long domain of RDV P2 has the inherent ability to recognize endosomal compartments even in the absence of viral infection. Thus, RDV P2–Rab5 interaction leads to the recruitment of Rab5 on the surface of virions within the exosomes. These observations are fully consistent with the role of RDV P2 in the viral package within MVBs via interaction with Rab5 and subsequent fusion of MVBs with the cell membrane for exosome secretion. Indeed, exosomes isolated from mammalian or plant cells have also been reported to contain the early endosome marker Rab5 (*Logozzi et al., 2009*; *Ramakrishnaiah et al., 2013*; *Song et al., 2019*). Purified exosomes secreted from infected leafhopper cells were larger and contained virions. The inhibition of exosomal secretion by silencing Rab27a or treatment with GW4869 suppresses such virus-mediated exosomal release, confirming the functional relevance of infectious exosomes with RDV dissemination (*Wei et al., 2008*; *Wei et al., 2009*). Thus, the use of the continuous insect cell culture system provides more conclusive proof that RDV can be released from insect vector cells via a Rab27a-dependent exosomal release pathway. Similarly, numerous arboviruses, such as DENV and ZIKV, are packaged in MVBs and secreted from cultured mosquito vector cells via the exosomal release pathway (*Martínez-Rojas et al., 2020*; *Reyes-Ruiz et al., 2019*; *Vora et al., 2018*). Thus, the exosomes play a conserved role in arbovirus dissemination.

Generally, in the secretory cells of insect salivary glands, arboviruses are frequently packaged in MVBs (*Ammar and Nault, 2002*; *Gray et al., 2014*; *Gray and Gildow, 2003*; *Janzen et al., 1970*; *Mao et al., 2017*; *Shikata and Maramorosch, 1965*). We thus anticipate that arboviruses may have evolved a conserved strategy to exploit the exosomal release pathway to ensure viral horizontal transmission from insect vectors into plant or mammalian hosts. In particular, exosome-mediated viral release from the apical plasmalemma into cavities does not cause substantial damage to salivary glands, conferring an evolved advantage for RDV to be persistently transmitted into plant phloem. Furthermore, exosomes potentially operate as an immune evasion strategy to ensure the initial infection of viruses in plant phloem (*Chahar et al., 2015*). Identification of the molecular modes or determinants of the transmission of arboviruses from vectors to hosts is a basic but very fundamental research objective. For the first time, we show the role of virus-induced exosomes in viral horizontal transmission from the salivary glands of an insect vector to a plant host in vivo.

Information concerning plant-cell–derived extracellular vesicles is rather limited. Most of the available information about plant exosomes highlights the implication of the SNARE SYP121 protein and comes from studies of plant–fungi interactions and plant–bacteria interactions (*Nielsen et al., 2012*; *Cai et al., 2018*). Rab11 is involved in exosome secretion and the recycling of cell wall components (*Nielsen et al., 2008*). The purified exosomes from *Arabidopsis* leaf apoplasts contained the Rab5 GTPase homologue ARA7 (*Rutter and Innes, 2018*). Turnip mosaic virus (TuMV) infection induces the formation of extracellular vesicles containing viral RNAs in infected leaves (*Movahed et al., 2019*). Rab proteins, such as RabG3E, RabD2B, and Rab7, were detected in TuMV-induced extracellular vesicles (*Movahed et al., 2019*). It is well known that various cellular or viral regulatory RNAs or proteins can be packaged into exosomes (*Chahar et al., 2015*), and thus, exosome-mediated viral horizontal transmission may modulate phloem–insect–virus interactions, an exciting frontier of plant science. Hence, our study shows that exosomes contribute to plant–insect interactions.

## Materials and methods

### Insects, viruses, and antibodies

Nonviruliferous individuals of leafhopper *N. cincticeps* were collected from Yunnan Province in southwestern China and propagated for several generations in the laboratory. Rice plants infected with RDV were also initially collected from Yunnan Province and propagated via transmission by *N.*

*cincticeps* under greenhouse conditions. Rabbit polyclonal antisera against RDV antigens and P8 were provided by Dr. Toshihiro Omura (National Agricultural Research Center, Japan). Polyclonal antibodies against Rab27a and Rab5 from *Drosophila* endosomes were obtained from Beyotime Biotechnology, Nantong, China. There were 57% and 79% amino acid sequence similarities between *Drosophila* and *N. cincticeps* Rab27a and between *Drosophila* and *N. cincticeps* Rab5, respectively (*Figure 5—figure supplement 3*). Preliminary tests also showed the specificity of these two antibodies in reacting with Rab27a or Rab5 of *N. cincticeps*. Polyclonal antibody against *N. cincticeps* CD63 (GenBank accession MW978787) was prepared by Genscript Biotech Corporation, Nanjing, China, as approved by the Science Technology Department of Jiangsu Province of China. IgGs were isolated from polyclonal antisera by using a protein A-Sepharose affinity column (Pierce Biotechnology, Waltham, MA, USA). IgGs against RDV antigens, Rab27a, CD63, or Rab5 were directly conjugated to FITC or rhodamine, respectively, according to the manufacturer's instructions (Thermo Fisher Scientific, Waltham, MA, USA). The actin dyes phalloidin-rhodamine and phalloidin-Alexa Fluor 647 carboxylic acid, His tag antibody conjugated to Alexa Fluor 488 (His-Alexa Fluor 488), and Myc tag antibody conjugated to Alexa Fluor 555 (Myc-Alexa Fluor 555) were obtained from Thermo Fisher Scientific. Actin was detected with actin-specific IgG (Sigma).

## Immunofluorescence microscopy for examination of viral infection and release in salivary glands

Nonviruliferous second-instar nymphs of *N. cincticeps* were fed on RDV-infected rice plants for 2 days and then maintained on healthy rice seedlings. At different days padp, the salivary glands of leafhoppers were dissected, fixed in 4% (v/v) paraformaldehyde in PBS for 2 hr, and then permeabilized in 0.2% Triton-X for 1 hr. The salivary glands were then immunolabeled with virus-specific IgG conjugated to FITC (virus-FITC), Rab27a-, CD63-, or Rab5-specific IgGs conjugated to rhodamine (Rab27a-rhodamine, CD63-rhodamine, or Rab5-rhodamine), and the actin dyes (phalloidin-rhodamine or phalloidin-Alexa Fluor 647 carboxylic acid). Immunostained salivary glands were then processed for immunofluorescence microscopy.

## Electron microscopy for examination of viral release into salivary cavities

At 14 days padp, salivary glands dissected from leafhoppers were fixed, dehydrated and embedded as previously described (*Mao et al., 2017*). The ultrathin sections were examined under a transmission electron microscope (H-7650; Hitachi, Tokyo, Japan). For immunoelectron microscopy, ultrathin sections were immunolabeled with virus-, Rab27a-, CD63-, or Rab5-specific IgG as the primary antibody and were then treated with goat anti-rabbit IgG conjugated with 15-nm-diameter gold particles as the secondary antibody (Abcam, Cambridge, UK).

## RT-qPCR and western blot assays to detect the effects of viral infection on exosomal production and release

At 14 days padp, total RNAs or proteins were extracted from the salivary glands of 30 viruliferous or nonviruliferous leafhoppers for to detect the relative transcript or protein levels of Rab27a and Rab5 during viral infection. RT-qPCR was performed using the SYBR Green PCR Master Mix kit (Promega, Madison, WI, USA). The actin transcript of *N. cincticeps* served as the internal reference, and the relative gene expression levels were calculated using the $2^{-\Delta\Delta CT}$ method. For western blot assay, RDV P8-, Rab27a-, CD63-, or Rab5-specific IgG served as the primary antibody and goat anti-rabbit IgG-peroxidase served as the secondary antibody. The band intensities of western blot assay were quantified with ImageJ software.

To examine the release of exosomes from salivary glands, more than 80 viruliferous or nonviruliferous leafhoppers prepared as described above were fed on two healthy rice seedlings (approximately 10 cm in height) for 2 days. The tested plants were then collected separately and subjected to protein extraction in equal quantities. Equal amounts of proteins from each group were loaded for western blot assay.

## Effect of synthesized dsRNAs on viral release from salivary glands

The dsRNAs targeting 500 bp regions of Rab27a or Rab5 were synthesized in vitro using the T7 RiboMAX Express RNAi System (Promega). Nonviruliferous second-instar nymphs were fed on diseased rice plants. At 8 days padp, the insects were microinjected with dsRab27a, dsRab5, or dsGFP (about 0.05 µg/insect) at the intersegment region of the thorax using a Nanoject II Auto-Nanoliter Injector (Spring). Thereafter, they were transferred to healthy rice seedling for recovery. At 14 days padp, the salivary glands of approximately 100 of these leafhoppers were first collected, and the corresponding bodies were individually tested using RT-PCR assay to confirm they were viruliferous. Then the salivary glands of dsRNAs-treated leafhoppers, of which bodies were viruliferous, were tested using RT-qPCR, western blot, or immunofluorescence assays to examine the effects of synthesized dsRNAs on the expression of RDV P8, Rab5, or Rab27a, as described above.

To investigate the transmission rates of RDV by *N. cincticeps* treated with dsRab27a, more than 300 s instar individuals were fed on RDV-infected rice plants for 2 days. At 8 days padp, groups of 100 insects were each microinjected with dsRab27a or dsGFP. At 14 days padp, the individual insects were transferred into glass tubes that each contained a single rice seedling for 2 days. Then the insects were individually tested using RT-PCR to determine whether they were viruliferous. Thirty days later, the plants inoculated with viruliferous *N. cincticeps* were subjected to RT-PCR detection of RDV P8 gene. The transmission rates of RDV by *N. cincticeps* were calculated as the percentage of RT-PCR-positive plants out of the total number of plants.

## Examination of RDV release from cultured leafhopper cells

Synchronous infection of continuous cultured cells of *N. cincticeps* with RDV was initiated as described by *Wei et al., 2006a*. At 48 hpi, infected cultured cells were fixed, dehydrated, and embedded. Then, ultrathin sections were cut and processed for electron microscopy, as previously described (*Wei et al., 2006a*). For immunoelectron microscopy, thin sections were immunolabeled with Rab27a- or CD63-specific IgG as the primary antibody and then treated with goat anti-rabbit IgG conjugated with 15-nm-diameter gold particles as the secondary antibody (Abcam).

To inhibit exosomal secretion from cultured cells via RNAi, 16 µg of dsRNAs and 12 µL of Cellfectin II reagent (Thermo Fisher Scientific) were diluted individually in 100 µL of LBM medium without antibiotics and fetal bovine serum, mixed gently together at room temperature for 20 min, and incubated with cultured cells for 8 hr. To chemically inhibit exosomal secretion, cultured cells were incubated with GW4869 (10 or 20 µM) or DMSO for 6 hr. Thereafter, cultured cells treated with dsRNAs, DMSO, or GW4869 were inoculated with purified RDV at a MOI of 1 in a solution of 0.1 M histidine that contained 0.01 M $MgCl_2$ (pH 6.2; His-Mg) at 25℃ for 2 hr, and then recovered with LBM medium. At 48 hpi, cultured cells were fixed, permeabilized, and immunolabeled with virus-specific IgG conjugated to rhodamine (virus-rhodamine), virus-FITC, or Rab27a-specific IgG conjugated to FITC (Rab27a-FITC) for immunofluorescence microscopy. Additionally, the treated cells were also harvested for RT-qPCR and western blot assays, as described above.

To investigate whether the inhibition of exosome formation affected viral replication, the cultured cells treated with dsRNAs, DMSO, or 20 µM GW4869 were inoculated with purified RDV at a MOI of 10. At 48 hpi, the extracellular medium and the cells were collected. The viral accumulation of each sample was determined by RT-qPCR assay, as described above.

## Exosome isolation

To isolate exosomes from the cellular supernatant, a Total Exosome Isolation kit (from cell culture media) (Thermo Fisher Scientific) was used. In brief, the media from infected or uninfected cultured leafhopper cells as well as dsRNAs-treated cells with 80% confluence were harvested and centrifuged at 2000 × g for 30 min to remove cells and debris. The samples were incubated with Exosome Isolation reagent at one half volume of the medium at 4℃ overnight. After centrifugation at 10,000 × g for 1 hr at 4℃, the pellets were resuspended. The exosomes were negatively stained with aqueous 2% uranyl acetate and then examined with an electron microscope.

## Yeast two-hybrid and GST pull-down assays

We used a Matchmaker Gold Yeast Two-hybrid system (Clontech, Mountain View, CA, USA) to examine the interaction of Rab27a or Rab5 with the outer capsid proteins of RDV. The Rab27a and

Rab5 genes were separately inserted into the pGADT7 vector to construct the prey plasmids. The full-length ORFs of P8, P2N (aa 1–688), and P2C (aa 689–1149) from RDV were separately cloned into the pGBKT7 vector as the bait plasmids and then used to transform yeast strain AH109 to confirm the absence of self-activation and toxicity. The prey and bait plasmids were then co-transformed into AH109, and β-galactosidase activity was detected on SD/-Ade-His-Leu-Trp/X-α-Gal culture medium. The interaction of pGBKT7-53 with pGADT7-T served as a positive control and that of pGBKT7-Lam with pGADT7-T served as a negative control.

A GST pull-down assay was conducted as described previously (*Chen et al., 2019*). The full-length ORFs of Rab5 and P8 were separately cloned into the pGEX-3x vector to construct plasmids expressing the GST fusion protein as baits (GST-Rab5 and GST-P8, respectively). Rab5 and P2N were separately cloned into the pEASY-Blunt E1 Expression Vector (Transgen Biotech, Beijing, China) to construct plasmids expressing the His fusion protein as preys (His-Rab5 and His-P2N). The recombinant proteins GST-Rab5 and GST-P8, as well as GST, were separately expressed in *Escherichia coli* stain BL21. Lysates were then incubated with glutathione-Sepharose beads (Amersham, Stafford, UK) and subsequently with the recombinant proteins His-P2N and His-Rab5, respectively. Finally, the eluates were analyzed by western blot assay using GST-tag and His-tag antibodies (Sigma-Aldrich), respectively.

## Recombinant baculovirus expressing P2N

The coding region of RDV P2N was amplified with a reverse primer that contained the coding sequences of the His-tag or GFP and a forward primer to construct P2N-His or P2N-GFP. The coding region of Rab5 was amplified with a reverse primer that contained the coding sequence of the Myc tag and a forward primer. These purified PCR products were cloned into the pDEST8 vector (Thermo Fisher Scientific), using an In-Fusion HD Cloning Kit (Clontech), to construct recombinant baculoviruses containing P2N or Rab5. Recombinant bacmids were generated by transforming *E. coli* DH10Bac (Thermo Fisher Scientific) with the recombinant baculoviruses.

Next, Sf9 cells were transfected with purified recombinant bacmid in the presence of Cellfectin II (Thermo Fisher Scientific), according to the manufacturer's instructions. At 48 hpi, the cells were processed for immunofluorescence microscopy with His-Alexa Fluor 488 or Rab5-rhodamine. Furthermore, cells expressing P2N-GFP were directly observed under a fluorescence microscope.

To chemically inhibit the exosomal pathway, Sf9 cells were incubated with GW4869 at a final concentration of 10 μM or 20 μM or chloroquine at a final concentration of 140 μM or 280 μM. At 48 hpi, treated Sf9 cells were fixed, permeabilized and immunolabeled with His-Alexa Fluor 488 for immunofluorescence microscopy. Then, the flow cytometry was performed to examine the effect of the inhibitory chemicals on the GFP fluorescence intensity of P2N-GFP. Briefly, approximately $1 \times 10^6$ cells were collected and washed once with PBS, resuspended in PBS, and immediately examined with a flow cytometer (Apogee, Hemel Hempstead, UK). Data from three independent biological experiments were analyzed using Histogram software and displayed as a plot of fluorescence intensity of GFP (*x*-axis) against cell number (*y*-axis).

## Frozen sections of rice plants

Then, approximately 15 viruliferous or nonviruliferous leafhoppers were reared in small cages to inoculate the two rice seedlings for 2 days, respectively (*Figure 7—figure supplement 1*). The feeding areas of the tested plants were embedded with O.C.T. Compound (Sakura, Torrance, CA, USA) and then sectioned with a Shandon Cryotome FSE (Thermo Fisher Scientific). The sections were ultimately immunolabeled with virus-FITC and Rab5-, CD63-, or Rab27a-rhodamine, and processed for immunofluorescence microscopy, as shown above. Rice without leafhopper feeding was treated exactly as the same.

## Statistical analyses

All quantitative data presented in the text and figures were analyzed with two-tailed *t*-tests in GraphPad Prism 7 (GraphPad Software, San Diego, CA, USA).

## Acknowledgements

We are grateful to Dr. Toshihiro Omura (National Agricultural Research Center, Tsukuba, Japan) for providing the antibodies against intact viral particles and P8. This work was supported by grants from the National Natural Science Foundation of China (Grants 31772124, 31920103014, and 31972239).

## Additional information

### Funding

| Funder | Grant reference number | Author |
|---|---|---|
| National Natural Science Foundation of China | 31772124 | Qian Chen |
| National Natural Science Foundation of China | 31920103014 | Taiyun Wei |
| National Natural Science Foundation of China | 31972239 | Qian Chen |

The funders had no role in study design, data collection and interpretation, or the decision to submit the work for publication.

### Author contributions

Qian Chen, Conceptualization, Formal analysis, Funding acquisition, Investigation, Visualization, Methodology, Writing - original draft, Project administration, Writing - review and editing; Yuyan Liu, Investigation, Visualization; Jiping Ren, Panpan Zhong, Manni Chen, Dongsheng Jia, Hongyan Chen, Investigation; Taiyun Wei, Conceptualization, Resources, Supervision, Visualization, Methodology, Project administration, Writing - review and editing

### Author ORCIDs

Qian Chen (ID) https://orcid.org/0000-0002-8442-6089
Taiyun Wei (ID) https://orcid.org/0000-0002-0732-9752

## Additional files

### Supplementary files

• Transparent reporting form

### Data availability

Sequencing data have been deposited in GenBank under accession numbers MW266984, MW266985 and MW978787.

The following datasets were generated:

| Author(s) | Year | Dataset title | Dataset URL | Database and Identifier |
|---|---|---|---|---|
| Chen Q, Liu Y, Ren J, Zhong P, Chen M, Jia D, Chen H, Wei T | 2021 | Exosomes mediate horizontal transmission of viral pathogens from insect vectors to plant phloem | https://www.ncbi.nlm.nih.gov/nuccore/MW978787 | NCBI Nucleotide, MW978787 |
| Chen Q, Liu Y, Ren J, Zhong P, Chen M, Jia D, Chen H, Wei T | 2020 | Exosomes mediate horizontal transmission of viral pathogens from insect vectors to plant phloem | https://www.ncbi.nlm.nih.gov/nuccore/MW266984 | NCBI Nucleotide, MW266984 |
| Chen Q, Liu Y, Ren J, Zhong P, Chen M, Jia D, Chen H, Wei T | 2020 | Exosomes mediate horizontal transmission of viral pathogens from insect vectors to plant phloem | https://www.ncbi.nlm.nih.gov/nuccore/MW266985 | NCBI Nucleotide, MW266985 |

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
