## [Decision Letter]

**Acceptance summary:**

Although the transmission of viruses by insects is well established, relatively little is known about what form they take when delivered to the plant phloem from insect stylets. This report demonstrates that rice dwarf virus is packaged into exosomes in its insect vector and that these exosomes are subsequently delivered to the plant phloem. These results have important implications for understanding virus transmission and movement with the plant. We think you have addressed the concerns raised previously and commend you on submitting a revised version that is significantly improved.

**Decision letter after peer review:**

Thank you for submitting your article "Exosomes mediate horizontal transmission of viral pathogens from insect vectors to plant phloem" for consideration by *eLife*. Your article has been reviewed by 3 peer reviewers, and the evaluation has been overseen by a Reviewing Editor and Detlef Weigel as the Senior Editor. The reviewers have opted to remain anonymous.

Essential revisions:

1) Verify exosome identify:

Some concerns were raised regarding the identity/origin of exosomes. These points are especially important to address as your main conclusion is that the virus is shuttled from the insect to the plant via insect-derived exosomes.

2) Antibody specificity:

Some concerns were raised regarding the specificity of antibodies used in immunofluorescence and western blot experiments. It is not clear that these antibodies have been properly vetted for use in this heterologous system. It is important to have controls without antibodies in some immunofluorescence assays to rule out any plant autofluorescence, as well as in western blots to rule out any cross-reaction with plant proteins.

3) Justify use of Rab markers for exosomes:

Some concerns were raised around the use of Rab GTPases as optimal markers for exosomes. The authors need to justify their use. Our suggestion is to conduct a few comparative assays alongside better-established exosomal markers (such as tetraspanins; orthologs of mammalian DC63).

4) Specificity of knock-downs in miRNA experiments:

Due to sequence similarity among Rab proteins, concerns were raised regarding the specificity of the miRNA-mediated knock down assays. Please verify that the intended transcripts have been specifically knocked down.

5) Size of exosomes:

Please provide exosome measurements to back up any claims of size differences.

*Reviewer #1:*

The authors used an arsenal of methods and experiments to show that the leafhopper transmitted RDV is associated with exosomes in the insect vector salivary system and it is secreted in the plant vascular system with the insect exosomes. the authors managed to show using electron and confocal microscopy that RDV is exosome-associated, however, controls in some experiments are missing, especially those showing the association of the insect exosomes with the plant vascular system.

This manuscript investigated the transmission of Rice Dwarf Virus (RDV) by leafhoppers and suggest that this virus uses the exosome pathway in salivary glands to be transmitted to host plant phloem by employing exosomal bodies which shuttle viral particles into the plant phloem where it starts infection. The manuscript presents an evidence of how vector-borne viruses use their relationships with their arthropod vectors to be well-transmitted to subsequent plant hosts and initiate infection. The use of exosome is a unique strategy by which membrane-associated structure are secreted into plant sieve elements for further infection and their formation is regulated by the expression of associated genes with the exosomal pathway. The authors used both electron and fluorescent microscopy as well as molecular biology tools to track the mode of transmission of RDV and the involved pathways. Some points require clarification and potentially additional experiments to fully support the conclusions.

1. In figure 1 the authors indicate the involvement of exosomes for shuttling RDV virions, however they did not use immunogold labeling as they did in figure 2 to track the virions. To support those images I would also perform immunogold labeling as in the size range of RDV and ribosomes and maybe other organelle structures are close and the images can be misleading without using specific antibody labeling and subsequent gold detection.

2. In figure 2 the authors indeed performed immunogold labeling, however, the amount of presented results in this regards are insufficient, and this kind of experiments needs to be quantified in order to make sure the labeling observed is not random as many times the gold particles attach non specifically. Although the subsequent results with regards to Rab27a, Rab5 and P8 show that those exosome-related genes are indeed induced upon virus infection and the silencing of two of them reduced their levels and colocalization with the virus.

3. I think better labeling of figure 4 panels, especially figure 4D and G is needed, to show the released virions which need to be marked.

4. Figure 7 shows that RDV is secreted into plant sieve elements with exosomes derived in the salivary glands of its leafhopper vector. The authors mainly used immunofluorescence and western blot. Control immunofluorescence images without antibodies or other controls are not provided. It is known that plant tissues strongly auto fluoresce at some wavelengths and thus it necessary to show controls for all these images.

5. The hypothesis is that exosome containing RDV virions are secreted into phloem. Since the authors can measure the diameter of such exosomes, I wonder if they can show measurements of exosomes and then compare this to the diameter of the salivary canal in the insect stylet. I assume the diameter of the salivary canal is way thinner than the exosome, and i wonder how such exosomes can move through the salivary canal if they are bigger. More information needs to be provided as this is the only route such exosomes would leave the insect and enter the plant.

*Reviewer #2:*

The authors present in this manuscript compelling data that the Rice Dwarf Virus (RDV) might be transmitted from the grasshopper vector into the phloem of rice host plant via insect exosomes. The authors support this hypothesis with their findings of i) co-localization of the vector MVB marker proteins Rab27a/Rab5 and virion particles and ii) physical interaction of the N-terminal viral capsid protein P2 (P2N) with vector Rab5. Moreover, transient knockdown of vector Rab27a/Rab5 via dsRNA treatment inhibits viral packaging into MVBs and rice transmission.

In general, the text is well written and experiments are well designed and explained. However, in order to state that exosomes are the means of virus transmission into rice phloem (see title) examination of exosomes with additional biomarkers and nanoparticle track analysis is necessary. Further, regarding the provided experimental data, some experiments would benefit from additional controls to ensure specificity of the experimental outcome and their interpretation.

Comments for the authors:

Rab27a and Rab5 dsRNA knockdown experiments. Please, provide control experiments of the specificity of targeted knockdown. Since Rab GTPases contain highly conserved domains, co-suppression might have occurred. For example, this could be tested by dsRNA Rab27a and Rab5 cross-check.

It was unclear whether Rab GTPases are optimal exosome biomarkers, and a suggestion is to include tetraspanins (orthologs of the mammalian CD63) as most commonly used exosome biomarkers.

It was unclear whether the specificity of the used antibodies has been vetted. Both anti-Rab antibodies are raised against *Drosophila* proteins and are polyclonal.

Quantification of difference in protein levels (Western blot) requires at least three biological replicates (Figure 2G, 3B, 4F, 5G, 7b).

In Figure 2A, the authors state that "Rab27a antibodies specifically labelled virus-containing exosomes", but Rab27a signals appear also in non-viral loaded exosomes in Figure 2A-II.

The authors state in lines 137/138"… KD Rab27a significantly reduced the accumulation of virus and Rab27a positive puncta in salivary glands". Such statement requires quantitative analysis and proper statistics in order to claim such significance.

Knockdown of Rab27a as well as chemical inhibition of endosome formation leads to reduced viral P8 expression (virus quantity) and cellular foci in fluorescence microscopy images. However, these data do not present exosomal release of virus, as interpreted by the authors.

The cell culture system was used to isolate exosomes and IGL via Rab27a antibody by electron microscopy. The authors claim that exosomes with viral particles are enlarged compared to non-virus containing exosomes. Here, size measurement and relative quantities by nanoparticle track analysis should be performed to support this statement. Additional biomarkers such as tetraspanins would support the identity of exosomes. Again, like in Figure 2, Rab27a labeled exosomes free of virus, thus Rab27a is not specific to virus-containing exosomes.

The authors state that "treatments of dsRNA Rab27a significantly abolished the formation of exosomes", similar to GW54869. Such statement must be verified by exosome quantification using nanoparticle track analysis.

No explanation was provided why Rab5 localization via antibody or Rab5-cMyc expression look so different, see Figure 6A versus B.

In Figure 6C, it seems that pictures for mock and DMSO have a longer exposure time than GW4869 and chloroquine. In order to claim any GFP signal intensity difference, quantitative analysis on biological replicates must be provided.

In Figure 7A, there seems to be cross-reaction of the Rab5 antibody with rice protein(s) leading to background signals in non-feeding samples. This makes it difficult to compare Rab5 translocation during feeding.

Figure 7D: Label WB with viruliferous or non- viruliferous instead of 1 and 2.

Here, I would suggest a complementary experiment, in which extracted exosomes, e.g. from infected grasshopper cell culture, to be injected in the rice phloem, to reproduce effects of viral spread, as observed in infected rice. Would such exosome injection lead to disease symptoms in rice?

Figure S1: Please clarify what Roman numbering indicates.

Figure S4B: What was the way to normalize fluorescence intensity (considering background noise)?

Figure S5E: axis labels of graphs are not readable.

*Reviewer #3:*

The manuscript under review addresses a timely and interesting question regarding how viruses are transmitted by insects and suggests that a plant virus, RDV, upon replicating in the cells of the insect vector, is secreted in exosomes and delivered to the insect saliva, and subsequently to the plant phloem. Although many studies have shown the involvement of exosomes and related machinery in animal viruses, very little is known about this phenomenon with plant viruses. The authors present very high-quality data in support of their conclusions. Indeed, the microscopy images are very well presented and convincing. At the same time, the authors present important data that suggest a very plausible mechanism for how the virus is packaged into multi-vesicular endosomes (MVEs) before being secreted. They show that the viral coat protein (CP) interacts with Rab5, a well-established identity marker for early endosomes. This mechanistic work is validated using several lines of evidence using protein-protein interaction methods and cell biology. Furthermore, they show that the CP and RDV particles associate with Rab27a, a marker for several secretion pathways, including exosomes. In addition, the association with the Rabs is shown to be biologically relevant as knock downs affect virus transmission. As such, there is a convincing story here regarding the packaging of RDV into exosomes in the insect vector.

There is only one major issue, regarding the putative detection of insect-derived exosomes in the plant that should either be addressed or be toned down (see point 9 below). Although the authors make a case for the insect proteins Rab5 and Rab27a being important for vectoring the virus, there are controls missing regarding the specificity of detection of these proteins in the plant. This point needs to be addressed.

The flow of the manuscript and presentation of data in the figures is good and relatively easy to follow. However, the writing needs to be significantly improved before publication in terms of form and grammar. The manuscript would also be improved by more discussion of how this data fits with the current state of knowledge on exosomes in plants. For example, Rab5 is associated with early endosomes. As such, is it logical, or consistent with the literature, to detect Rab5 in exosomes? This report will be of interest to a broad range of researchers and it is important to introduce and discuss the dynamics of the endomembrane system to this broad audience.

Comments for the authors:

1. Figure 3b. The RNAi-mediated knock-down of Rab27b does not result in a dramatic reduction of the Rab27b protein (1 vs. 1.41). Is this result representative? That is, the qRT-PCR results suggest that the efficacy of knock down varies widely between individual insects.

2. At several points, the authors indicate that virus-containing exosomes as being larger than normal exosomes, but is there some quantification for this?

3. Figure legends: generally, the figure legends are not acceptable for a scientific publication. Legends need to contain all the information required to understand what was done in the experiments, without interpretation of the results. For example, multiple figure legends talk about immunostaining with virus-rhodamine red and Rab27a-FITC. Presumably, the virus was detected with anti virus antibodies labelled with rhodamine, not with actual virus. Likewise, in Figure 5C it is simply indicated "GST pulldown" or "proteins incubated with cell lysate". While one can figure out what is meant, this is not the way to formally write a figure legend. These are just two small examples and all legends should be revisited.

4. Figure 4e. KD of Rab27a is again not that remarkable by western. At the same time neither is reduction of virus P8 – 1.39 in dsGFP vs. 1 in dsRab27a (would it not make more sense to normalize to dsGFP as 1).

However, would it not actually make sense that virus levels are not that different upon KD of Rab27a? According to the authors, Rab27a is important for virus transmission, but it should not affect replication of the virus; only its ability to infect other cells. Presumably, replication is not taking place in the MVBs.

Same point for results using GW4869.

5. Same point as above for Figure 5 and KD or Rab5. It is perhaps not feasible, but would it be more informative to investigate difference in total virus accumulation versus extracellular virus accumulation (in cell culture – maybe in saliva?).

6. Figure 5D (and elsewhere): The images indicate exosomes and virus-containing exosomes. The latter are from virus-infected cells, but in this case, how do you know that there are virus particles in these exosomes? Is there a method used to differentiate the "bodies" in regular exosomes versus viruses"

7. Figure 6B: P2N shows colocalization with Rab5 and, in a separate set of experiments, with Rab27a. Thus, P2N colocalizes with both proteins, but are all three colocalized. In vertebrate systems invagination of exosomes is thought to be concomitant to replacement of Rab5 with Rab7, presumably prior to recruitment of Rab27a, which is associated with later stages related to secretion. Given what is known about these Rab GTPases, is there precedent for them to be present on the same vesicles.

Do you have direct data showing colocalization of Rab5 and Rab27a?

8. Please discuss more what is known about exosomes in plants and if possible their known associations with different Rabs.

9. Figure 7D. Western blots show Rab5 and Rab27a accumulation in plants exposed to leafhoppers, with a moderate difference in accumulation of both when exposed to viruliferous insects. These blots should include samples from plants not exposed to leafhoppers to demonstrate a lack of cross-reaction to any plant protein. This is particularly important given that there is a non-negligible background in the immunofluorescence microscopy of non-feeding plants. The antibodies used in this study were raised against *Drosophila* proteins and quick BLAST of *Drosophila* Rab5, and to a lesser extent Rab27a, against rice shows significant blocks of exact or near exact homology.

It is thus important to determine if the signals seen are Rab GTPases from the insect or from the plant. It is not implausible that feeding from viruliferous and non-viruliferous could induce the expression of Rab GTPases. Indeed, given the amount Rab proteins putatively transferred from insect to plant, these proteins would presumably be diluted throughout the entire plant. Is it plausible that the insect would secrete enough of these proteins to detectable in all parts of the phloem?

10. Supplementary figure 4 (and related to figure 6): data is shown on the quantification of fluorescence as a measure of P2N. It is indicated that there is less in the presence of the two inhibitors. However, if there is no longer a concentration of P2N in specific vesicles, then one would expect the diffuse protein to be much less detectable. As such, and quantification based on fluorescence is probably not accurate and should rather be assayed by western blot.

11. Movahed et al., have reported Turnip Mosaic Virus Components Are Released into the Extracellular Space by Vesicles in Infected Leaves. These vesicles also appeared to contain several Rab proteins, albeit not the as those investigated here (and these are plant derived exosomes). Nonetheless, this study should be discussed in the context of the current manuscript.

---

## [Author Response]

Essential revisions:1) Verify exosome identify:Some concerns were raised regarding the identity/origin of exosomes. These points are especially important to address as your main conclusion is that the virus is shuttled from the insect to the plant via insect-derived exosomes.

We prepared for the antibody against CD63 of *N. cincticeps*, which can be used for labelling the MVB and exosomes in the salivary glands and cultured leafhopper cells. The related contents appeared in Figures 2, 4 and 7. Furthermore, we have added the necessary plant or antibody controls to prove the specificity for Rab5, Rab27a and CD63 antibodies to react with the exosome components secreted from insect vectors, but not with rice components. Related results were shown in Figure 7.

2) Antibody specificity:Some concerns were raised regarding the specificity of antibodies used in immunofluorescence and western blot experiments. It is not clear that these antibodies have been properly vetted for use in this heterologous system. It is important to have controls without antibodies in some immunofluorescence assays to rule out any plant autofluorescence, as well as in western blots to rule out any cross-reaction with plant proteins.

The amino acid sequences of Rab27a and Rab5 of *N. cincticeps* had no significant similarity by using Blast N, although they showed the conserved domain of Rab GTPases. The relative expression of Rab27a and Rab5 genes of *N. cincticeps* in the salivary glands after the treatment with dsRab27a or Rab5 were tested. The absence of co-suppression in the treatments of dsRab27a or dsRab5 was concluded. Furthermore, there were 57% and 79% amino acid sequence similarities between *Drosophila* and *N. cincticeps* Rab27a and between *Drosophila* and *N. cincticeps* Rab5, respectively. Although both Rab antibodies were prepared against the antigen derived from *Drosophila*, these antibodies showed the specificity of *N. cincticeps* samples in western blots with correct molecular weights, and in the immunofluorescence and immunoelectron microscopy with the localization of MVBs and exosomes. These results confirmed that both Rab antibodies in this study were useful for recognizing exosomes in *N. cincticeps*. The related contents were shown on lines 241-246, 389-391, Figures 2-5 and Figure 5—figure supplement 2.

We have added the necessary plant or antibody controls to prove the specificity for Rab5, Rab27a and CD63 antibodies to react with the exosome components secreted from insect vectors, but not with rice components. The controls without antibodies were added to exclude the plant autofluorescence. Western blot assay showed that the utilization of Rab5, Rab27a and CD63 antibodies did not cause the cross-reaction with plant proteins. The related contents were shown in Figures 7.

3) Justify use of Rab markers for exosomes:Some concerns were raised around the use of Rab GTPases as optimal markers for exosomes. The authors need to justify their use. Our suggestion is to conduct a few comparative assays alongside better-established exosomal markers (such as tetraspanins; orthologs of mammalian DC63).

We prepared for the antibody against CD63 of *N. cincticeps*, which can be used for labelling the MVB and exosomes in the salivary glands and leafhopper cultured cells. The related contents were shown in Figures 2, 4 and 7.

4) Specificity of knock-downs in miRNA experiments:Due to sequence similarity among Rab proteins, concerns were raised regarding the specificity of the miRNA-mediated knock down assays. Please verify that the intended transcripts have been specifically knocked down.

The amino acid sequences of Rab27a and Rab5 of *N. cincticeps* had no significant similarity by using Blast N, although they showed the conserved domain of Rab GTPases. The relative expression of Rab27a and Rab5 genes of *N. cincticeps* in the salivary glands after the treatment with dsRab27a or Rab5 was tested. RT-qPCR assay also showed that the knockdown of Rab5 expression did not significantly affect Rab27a expression, and that the knockdown of Rab27a expression also did not significantly affect Rab5 expression. The related contents were shown in Figure 5—figure supplement 2D.

5) Size of exosomes:Please provide exosome measurements to back up any claims of size differences.

We have measured the sizes of virus-free and virus-containing exosomes by negative staining electron microscopy, and showed that the virus-containing exosomes are larger than virus-free exosomes. The related contents were shown in Figure 1K and Figure 4D.

Reviewer #1:[…] 1. In figure 1 the authors indicate the involvement of exosomes for shuttling RDV virions, however they did not use immunogold labeling as they did in figure 2 to track the virions. To support those images I would also perform immunogold labeling as in the size range of RDV and ribosomes and maybe other organelle structures are close and the images can be misleading without using specific antibody labeling and subsequent gold detection.

We characterized double-layered RDV particles of approximately 65 nm in diameter, which were bigger than ribosomes with a diameter of 25-30 nm. We have performed immunogold labeling to track the virions. The related results were shown on lines 119-121 and 172-173, as well as Figure 1—figure supplement 2.

2. In figure 2 the authors indeed performed immunogold labeling, however, the amount of presented results in this regards are insufficient, and this kind of experiments needs to be quantified in order to make sure the labeling observed is not random as many times the gold particles attach non-specifically. Although the subsequent results with regards to Rab27a, Rab5 and P8 show that those exosome-related genes are indeed induced upon virus infection and the silencing of two of them reduced their levels and colocalization wit the virus.

We have presented the new images in Figure 2A-B to show the immunogold labelling of Rab27a and CD63 on the MVBs or exosomes. For these immunoelectron microscopic images, the gold particles on MVBs or exosomes are specific and not random.

3. I think better labeling of figure 4 panels, especially figure 4D and G is needed, to show the released virions which need to be marked.

The arrows in the enlarge images and the new images in Figure 4E and F indicated the released virions.

4. Figure 7 shows that RDV is secreted into plant sieve elements with exosomes derived in the salivary glands of it's leafhopper vector. The authors mainly used immunofluorescence and western blot. Control immunofluorescence images without antibodies or other controls are not provided. It is known that plant tissues strongly auto fluoresce at some wavelengths and thus it necessary to show controls for all these images.

The controls without antibodies were added to Figure 7A to show the weak background caused by autofluorescence of plant.

5. The hypothesis is that exosome containing RDV virions are secreted into phloem. Since the authors can measure the diameter of such exosomes, I wonder if they can show measurements of exosomes and then compare this to the diameter of the salivary canal in the insect stylet. I assume the diameter of the salivary canal is way thinner than the exosome, and i wonder how such exosomes can move through the salivary canal if they are bigger. More information needs to be provided as this is the only route such exosomes would leave the insect and enter the plant.

The average diameters of salivary canal of most leafhoppers in Cicadellidae range from 3 μm to 500 nm (Brozek and Herczek, 2000; Zhao et al., 2010; Leopold et al., 2003), which are bigger than the virus-containing exosome. Therefore, such virus-containing exosomes can move through the salivary canal. We also added this discussion on lines 306-309.

Reviewer #2:[…] Comments for the authors:Rab27a and Rab5 dsRNA knockdown experiments. Please, provide control experiments of the specificity of targeted knockdown. Since Rab GTPases contain highly conserved domains, co-suppression might have occurred. For example, this could be tested by dsRNA Rab27a and Rab5 cross-check.

The amino acid sequences of Rab27a and Rab5 of *N. cincticeps* had no significant similarity according to Blast N, although they showed the conserved domain of Rab GTPases. The relative expression of Rab27a and Rab5 genes of *N. cincticeps* in the salivary glands after the treatment with dsRab27a or Rab5 was tested. RT-qPCR assay also showed that the knockdown of Rab5 expression did not significantly affect Rab27a expression and that the knockdown of Rab27a expression also did not significantly affect Rab5 expression. The related contents were shown in Figure 5—figure supplement 2D.

It was unclear whether Rab GTPases are optimal exosome biomarkers, and a suggestion is to include tetraspanins (orthologs of the mammalian CD63) as most commonly used exosome biomarkers.

We have prepared for the antibody against CD63 of *N. cincticeps*, which can be used for labelling the MVB and exosomes in the salivary glands and leafhopper cultured cells. The related contents were shown in Figures 2, 4 and 7.

It was unclear whether the specificity of the used antibodies has been vetted. Both anti-Rab antibodies are raised against Drosophila proteins and are polyclonal.

Rab27a or Rab5 from *N. cincticeps* and *Drosophila* had 57% or 79% amino acid similarity, respectively. Although both Rab antibodies were prepared against the antigen derived from *Drosophila*, these antibodies showed the specificity of *N. cincticeps* samples in western blots with correct molecular weights, and in the immunofluorescence and immunoelectron microscopy with the localization of MVBs and exosomes. These results confirmed that both Rab antibodies in this study were useful for recognizing exosomes in *N. cincticeps*. The related contents were shown on lines 244-246, 389-391, and on Figures 2-5 and Figure 5—figure supplement 2E.

Quantification of difference in protein levels (Western blot) requires at least three biological replicates (Figure 2G, 3B, 4F, 5G, 7b).

The results of western blot represent three biological replicates. The descriptions were shown in the Figure Legends of Figures 2G, 3B, 4I, 5H, 5J, 7D and Figure 6—figure supplement 1C.

In Figure 2A, the authors state that "Rab27a antibodies specifically labelled virus-containing exosomes", but Rab27a signals appear also in non-viral loaded exosomes in Figure 2A-II.

We have revised the similar sentences in the full text.

The authors state in lines 137/138"… KD Rab27a significantly reduced the accumulation of virus and Rab27a positive puncta in salivary glands". Such statement requires quantitative analysis and proper statistics in order to claim such significance.

We have added the results of quantitative analysis to show the change of virus, Rab5, Rab27a and CD63, as detected by immunofluorescence microscopy in Figures 2E, 3D, and 5F.

Knockdown of Rab27a as well as chemical inhibition of endosome formation leads to reduced viral P8 expression (virus quantity) and cellular foci in fluorescence microscopy images. However, these data do not present exosomal release of virus, as interpreted by the authors.

To further confirm that the treatment of dsRab27a and GW4869 would inhibit viral exosomal release from infected cultured cells without significant effect on viral propagation, we perform one new experiment. The cultured cells treated with dsRNAs (dsGFP and dsRab27a), DMSO or GW4869 were inoculated with purified RDV at a MOI of 10 which guaranteed viral infection rate was 100%. At 48 hpi, the extracellulaar medium and the cells were collected. We showed that the treatment of dsRab27a or GW4869 significantly reduced RDV P8 transcript loads in the extracellular medium, but did not significantly reduce RDV P8 transcript loads in cell-associated viruses. These results demonstrated that RDV had proliferated in the infected cells but viral release from the cells had been impeded by the inhibition of exosomes formation. The related contents were shown on lines 192-203 and Figure 4—figure supplement 1.

The cell culture system was used to isolate exosomes and IGL via Rab27a antibody by electron microscopy. The authors claim that exosomes with viral particles are enlarged compared to non-virus containing exosomes. Here, size measurement and relative quantities by nanoparticle track analysis should be performed to support this statement. Additional biomarkers such as tetraspanins would support the identity of exosomes. Again, like in Figure 2, Rab27a labeled exosomes free of virus, thus Rab27a is not specific to virus-containing exosomes.

For the size measurement and relative quantities by nanoparticle track analysis, we have contacted the expert of this method, and known that the quantity of purified exosomes from the salivary glands or medium of infected monolayer of cultured leafhopper cells are not enough for this analysis. Thus, we have measured the sizes of virus-free and virus-containing exosomes in salivary glands, and proved the virus-containing exosomes were larger compared to virus-free exosomes. We also used negative staining electron microscopy to observe the purified exosomes from cultured cells and analyzed the size. The related contents were shown on lines 121-123 and 177-180, as well as Figures 1K and 4D.

We prepared for the antibody against CD63 of *N. cincticeps*, which can be used for labelling the MVB and exosomes in the salivary glands and leafhopper cultured cells. The related contents were shown in Figures 2, 4 and 7. For the comment “Again, like in Figure 2, Rab27a labeled exosomes free of virus, thus Rab27a is not specific to virus-containing exosomes”, we have checked the full text and revised the similar sentences for Rab27a, CD63 and Rab5.

The authors state that "treatments of dsRNA Rab27a significantly abolished the formation of exosomes", similar to GW54869. Such statement must be verified by exosome quantification using nanoparticle track analysis.

For the relative quantities by nanoparticle track analysis, we have contacted the expert of this method, and known that the quantity of purified exosomes from the medium of infected monolayer of *N. cincticeps* cultured cells are not enough for this analysis. Instead, we performed the negative staining electron microscopy to analyze the impaired purified exosomes secreted from dsRNAs- and GW4869-treated cells. The related contents were shown on lines 184-187, and Figure 4G.

No explanation was provided why Rab5 localization via antibody or Rab5-cMyc expression look so different, see Figure 6A versus B.

We used the baculovirus expression system in Sf9 cells to overexpress *N. cincticeps* Rab5 fused with cMyc tag (Rab5-cMyc). The results showed that the expressed *N. cincticeps* Rab5 colocalized with RDV P2N fused with His (Figure 6A). Furthermore, we also used antibody against *Drosophila* Rab5 to label Rab5 of Sf9 cells. The results showed RDV P2N could colocalize with Rab5 of Sf9 cells (Figure 6B). The explanation was on lines 248-258.

In Figure 6C, it seems that pictures for mock and DMSO have a longer exposure time than GW4869 and chloroquine. In order to claim any GFP signal intensity difference, quantitative analysis on biological replicates must be provided.

We have provided the new images for mock and DMSO, which did not show a longer exposure time than GW4869 and chloroquine in Figure 6C. Furthermore, we have quantified the GFP signal intensity of mock, DMSO, GW4869 and chloroquine treatments, and did three biological replicates. The results were demonstrated in Figure 6—figure supplement 1 and 2.

In Figure 7A, there seems to be cross-reaction of the Rab5 antibody with rice protein(s) leading to background signals in non-feeding samples. This makes it difficult to compare Rab5 translocation during feeding.

Rab5 antibody did cause weak background signal in non-feeding samples in some degree. However, with the feeding time increasing, the immunofluorescence of Rab5 antibody in samples of nonviruliferous leafhopper feeding or viruliferous leafhopper feeding strengthened, and showed obvious differences. These results showed that Rab5 translocation to plant phloem during vector feeding. We also added a control without antibodies to show the autofluorescence of plant tissues. We hope these data could prove the release of Rab5 from leafhopper to rice plants. The results were shown in Figure 7A.

I am wondering, if it would be possible to isolate exosomes from salivary glands from leafhoppers either non-viruliferous, viruliferous, or viruliferous upon injection of dsRNA Rab27a. Such data would significantly support the engulfment of virus into exosomes complementing the here provided EM images.

It seems impossible for us to isolate exosomes from salivary glands from leafhoppers, because the leafhoppers are so tiny that the exosome isolated from salivary glands using the general method are very limited.

Figure 7D: Label WB with viruliferous or non- viruliferous instead of 1 and 2.

We have revised this.

Here, I would suggest a complementary experiment, in which extracted exosomes, e.g. from infected grasshopper cell culture, to be injected in the rice phloem, to reproduce effects of viral spread, as observed in infected rice. Would such exosome injection lead to disease symptoms in rice?

This suggestion is good for the inoculation of animal cultured cells with exosomes purified from insect cultured cells. However, as a persistent-propagative plant virus (RDV), the purified virus-containing exosomes from infected cultured cells of leafhopper must be firstly acquired by insect vectors during feeding, and then the viruses can transmit into plants from insect salivary glands. Thus, it is impossible for the direct inoculation of purified virus-containing exosomes from infected cultured cells into rice plants.

Figure S1: Please clarify what Roman numbering indicates.

We have explained this in Figure legend.

Figure S4B: What was the way to normalize fluorescence intensity (considering background noise)?

We further used western blot assay to test the expression levels of P2N-His in different treatments in Figure 6—figure supplement 1C.

Figure S5E: axis labels of graphs are not readable.

We are sorry for making you puzzled. We added the explanations to the Figure legend.

Reviewer #3:The manuscript under review addresses a timely and interesting question regarding how viruses are transmitted by insects and suggests that a plant virus, RDV, upon replicating in the cells of the insect vector, is secreted in exosomes and delivered to the insect saliva, and subsequently to the plant phloem. Although many studies have shown the involvement of exosomes and related machinery in animal viruses, very little is known about this phenomenon with plant viruses. The authors present very high-quality data in support of their conclusions. Indeed, the microscopy images are very well presented and convincing. At the same time, the authors present important data that suggest a very plausible mechanism for how the virus is packaged into multi-vesicular endosomes (MVEs) before being secreted. They show that the viral coat protein (CP) interacts with Rab5, a well-established identity marker for early endosomes. This mechanistic work is validated using several lines of evidence using protein-protein interaction methods and cell biology. Furthermore, they show that the CP and RDV particles associate with Rab27a, a marker for several secretion pathways, including exosomes. In addition, the association with the Rabs is shown to be biologically relevant as knock downs affect virus transmission. As such, there is a convincing story here regarding the packaging of RDV into exosomes in the insect vector.There is only one major issue, regarding the putative detection of insect-derived exosomes in the plant that should either be addressed or be toned down (see point 9 below). Although the authors make a case for the insect proteins Rab5 and Rab27a being important for vectoring the virus, there are controls missing regarding the specificity of detection of these proteins in the plant. This point needs to be addressed.

We have added the necessary plant or antibody controls to prove the specificity for antibodies against Rab5, Rab27a and CD63 to react with exosome components secreted from insect vectors, but not with rice components. Related results were shown in Figure 7.

The flow of the manuscript and presentation of data in the figures is good and relatively easy to follow. However, the writing needs to be significantly improved before publication in terms of form and grammar. The manuscript would also be improved by more discussion of how this data fits with the current state of knowledge on exosomes in plants. For example, Rab5 is associated with early endosomes. As such, is it logical, or consistent with the literature, to detect Rab5 in exosomes? This report will be of interest to a broad range of researchers and it is important to introduce and discuss the dynamics of the endomembrane system to this broad audience.

In the Introduction and Discussion, we have added the sentences to induce and discuss the dynamics of the exosomes biogenesis, including plant system. The published literatures indicated that Rab5 was also detected in the exosomes or extracellular vesicles isolated from mammalian or plant cells, especially in virus-infected systems. Furthermore, our data of RDV P2-Rab5 interaction would lead to the recruitment of Rab5 on the surface of virions within the exosomes. These observations (Figure 5) are fully consistent with the role of RDV P2 in viral package within MVB via interaction with Rab5. Thus, our finding is logical and consistent with the published literatures. The related contents appear on lines 65-72 in the Introduction, and on lines 302-306, 332-338 in the Discussion.

Comments for the authors:1. Figure 3b. The RNAi-mediated knock-down of Rab27b does not result in a dramatic reduction of the Rab27b protein (1 vs. 1.41). Is this result representative? That is, the qRT-PCR results suggest that the efficacy of knock down varies widely between individual insects.

We have performed the western blot again, and analyzed the relative intensities of bands. The new data was shown in Figure 3B.

2. At several points, the authors indicate that virus-containing exosomes as being larger than normal exosomes, but is there some quantification for this?

We have measured the sizes of virus-free and virus-containing exosomes, and proved that virus-containing exosomes were larger compared to virus-free exosomes. The related sentences were added on lines 121-123 and 177-180. The images were added to Figures 1K and 4D.

3. Figure legends: generally, the figure legends are not acceptable for a scientific publication. Legends need to contain all the information required to understand what was done in the experiments, without interpretation of the results. For example, multiple figure legends talk about immunostaining with virus-rhodamine red and Rab27a-FITC. Presumably, the virus was detected with anti virus antibodies labelled with rhodamine, not with actual virus. Likewise, in Figure 5C it is simply indicated "GST pulldown" or "proteins incubated with cell lysate". While one can figure out what is meant, this is not the way to formally write a figure legend. These are just two small examples and all legends should be revisited.

We have carefully revised the Figure legends in the full text.

4. Figure 4e. KD of Rab27a is again not that remarkable by western. At the same time neither is reduction of virus P8 – 1.39 in dsGFP vs. 1 in dsRab27a (would it not make more sense to normalize to dsGFP as 1).

Thank you very much. We analyzed the intensities of bands again, and normalize to dsGFP as 1. The reduction of Rab27a is remarkable: 0.43 in dsRab27a vs. 1.00 in dsGFP. The reduction of P8 0.54 in dsRab27a vs. 1.00 in dsGFP. The relative data was also renewed in Figure 4I.

However, would it not actually make sense that virus levels are not that different upon KD of Rab27a? According to the authors, Rab27a is important for virus transmission, but it should not affect replication of the virus; only its ability to infect other cells. Presumably, replication is not taking place in the MVBs. Same point for results using GW4869.

To further confirm that the treatment of dsRab27a and GW4869 would inhibit viral exosomal release from infected cultured cells without significant effect on viral propagation, we perform one new experiment. The cultured cells treated with dsRNAs (dsGFP and dsRab27a), DMSO or GW4869 were inoculated with purified RDV at a MOI of 10 which guaranteed viral infection rate was 100%. At 48 hpi, the extracellular medium and the cells were collected. We showed that the treatment of dsRab27a or GW4869 significantly reduced RDV P8 transcript loads in the extracellular medium, but did not significantly reduce RDV P8 transcript loads in cell-associated viruses. These results demonstrated that RDV had proliferated in the infected cells but viral release from the cells had been impeded by the inhibition of exosomes formation. The related contents were shown on lines 192-203 and Figure 4—figure supplement 1.

5. Same point as above for Figure 5 and KD or Rab5. It is perhaps not feasible, but would it be more informative to investigate difference in total virus accumulation versus extracellular virus accumulation (in cell culture – maybe in saliva?).

We have analyzed the accumulation levels of cell-associated viruses and extracellular viruses in leafhopper cultured cells after different treatments for inhibiting exosomal formation. However, such experiments are not available for insect saliva due to the limited harvest.

6. Figure 5D (and elsewhere): The images indicate exosomes and virus-containing exosomes. The latter are from virus-infected cells, but in this case, how do you know that there are virus particles in these exosomes? Is there a method used to differentiate the "bodies" in regular exosomes versus viruses"

We added the evidence of immunoelectron microscopy to prove the double-layered particles in exosomes were viruses. RDV particles, confirmed by immunoelectron microscopy, were easily distinguished by their spherical appearance and diameter (65 nm), and they were not found in uninfected controls. The related contents were shown on lines 111-114, 119-121, 172-173 and Figure 1—figure supplement 2.

7. Figure 6B: P2N shows colocalization with Rab5 and, in a separate set of experiments, with Rab27a. Thus, P2N colocalizes with both proteins, but are all three colocalized. In vertebrate systems invagination of exosomes is thought to be concomitant to replacement of Rab5 with Rab7, presumably prior to recruitment of Rab27a, which is associated with later stages related to secretion. Given what is known about these Rab GTPases, is there precedent for them to be present on the same vesicles. Do you have direct data showing colocalization of Rab5 and Rab27a?

We have added Figure 5—figure supplement 2E to show that some puncta labeled by Rab27a-FITC could colocalize with the puncta labeled by Rab5-rhodamine in the salivary glands from nonviruliferous *N. cincticeps*.

Furthermore, our data of RDV P2-Rab5 interaction would lead to the recruitment of Rab5 on the surface of virions within the exosomes. These observations (Figure 5) are fully consistent with the role of RDV P2 in viral package within MVB via interaction with Rab5. In the Introduction and Discussion, we have added the sentences to discuss the dynamics of the exosomes biogenesis, including plant system. The published literatures indicated that Rab5 was also detected in the exosomes or extracellular vesicles isolated from mammalian or plant cells, especially in virus-infected systems. Thus, our finding is logical and consistent with the published literatures. The related contents appear on lines 65-72 in the Introduction, and on lines 302-306, 332-338 in the Discussion.

8. Please discuss more what is known about exosomes in plants and if possible their known associations with different Rabs.

We have discussed the exosomes in plants and their known associations with Rabs on lines 364-373.

9. Figure 7D. Western blots show Rab5 and Rab27a accumulation in plants exposed to leafhoppers, with a moderate difference in accumulation of both when exposed to viruliferous insects. These blots should include samples from plants not exposed to leafhoppers to demonstrate a lack of crossreaction to any plant protein. This is particularly important given that there is a non-negligible background in the immunofluorescence microscopy of non-feeding plants. The antibodies used in this study were raised against Drosophila proteins and quick BLAST of Drosophila Rab5, and to a lesser extent Rab27a, against rice shows significant blocks of exact or near exact homology.

We reconducted this biological experiment for 3 replicates. Western blot assay proved the obvious difference of Rab5, CD63 or Rab27a accumulation between viruliferous and non-viruliferous feeding samples. Furthermore, the utilization of Rab5, CD63 or Rab27a antibody did not cause the cross-reaction with plant proteins. Related results were shown in Figure 7D.

The amino acid sequences of Rab27a and Rab5 of *N. cincticeps* had no significant similarity according to Blast N, although they showed the conserved domain of Rab GTPases. The relative expression of Rab27a and Rab5 genes of *N. cincticeps* in the salivary glands treated with dsRab27a or Rab5 were tested. The absence of co-suppression with the treatments of dsRab27a or dsRab5 was concluded. The related results were shown in Figure 5—figure supplement 2D.

Furthermore, there were 57% and 79% amino acid sequence similarities between *Drosophila* and *N. cincticeps* Rab27a and between *Drosophila* and *N. cincticeps* Rab5, respectively. Although both Rab antibodies were prepared against the antigen derived from *Drosophila*, these antibodies showed the specificity of *N. cincticeps* samples in western blot with correct molecular weights, and in the immunofluorescence and immunoelectron microscopy with the localization of MVBs and exosomes. These results confirmed that both Rab antibodies in this study were useful for recognizing exosomes in *N. cincticeps*. The related contents were shown on lines 241-246, 389-391, and on Figures 2-5 and Figure 5—figure supplement 3.

It is thus important to determine if the signals seen are Rab GTPases from the insect or from the plant. It is not implausible that feeding from viruliferous and non-viruliferous could induce the expression of Rab GTPases. Indeed, given the amount Rab proteins putatively transferred from insect to plant, these proteins would presumably be diluted throughout the entire plant. Is it plausible that the insect would secrete enough of these proteins to detectable in all parts of the phloem?

We have added the plant controls which were not fed on by leafhoppers. The immunofluorescence result showed the weak background signals of Rab27a and Rab5 in the non-feeding control, and showed strong and specific fluorescence signals in leafhopper-feeding samples. It was suggested that the signals were from insect but not plant. Furthermore, western blot assay confirmed that Rab27a and Rab5 antibodies did not react with the plant samples. These results were shown in Figure 7A and 7D.

Theoretically, Rab proteins would be diluted throughout the entire plant. However, as shown in Figure 7—figure supplement 1, in our study, approximate 15 viruliferous or nonviruliferous leafhoppers were reared in small cages to inoculate rice seedlings for 2 days. We even fed 80 viruliferous or nonviruliferous leafhoppers on 2 rice seedlings to enrich the proteins for western blot assay. Such methods are useful for accumulation of enough amount Rab proteins putatively transferred from insect to plant phloem. The related results were shown in Figure 7.

10. Supplementary figure 4 (and related to figure 6): data is shown on the quantification of fluorescence as a measure of P2N. It is indicated that there is less in the presence of the two inhibitors. However, if there is no longer a concentration of P2N in specific vesicles, then one would expect the diffuse protein to be much less detectable. As such, and quantification based on fluorescence is probably not accurate and should rather be assayed by western blot.

We have added western blot to show the difference of P2N expression level among the treatments in Figure 6—figure supplement 1C. Furthermore, in Figure 6—figure supplement 2, we carefully compared the quantification of fluorescence as a measure of P2N-GFP after the treatment of inhibitors.

11. Movahed et al., have reported Turnip Mosaic Virus Components Are Released into the Extracellular Space by Vesicles in Infected Leaves. These vesicles also appeared to contain several Rab proteins, albeit not the as those investigated here (and these are plant derived exosomes). Nonetheless, this study should be discussed in the context of the current manuscript.

We have discussed the related contents of TuMV extracellular vesicles and their associated Rab proteins on lines 370-373.